# Visual Prompting with Iterative Refinement for Design Critique Generation

## Abstract

Feedback is essential in all design processes, such as user interface (UI) design. Automating design critiques can significantly enhance design workflow efficiency. Although existing vision language models (VLMs) excel in many tasks, they often struggle with generating high-quality design critiques—a complex task that requires producing detailed design comments that are visually grounded in a given design's image. Building on recent advancements in iterative refinement of text output and visual prompting methods, we propose a multimodal iterative refinement and visual prompting framework for UI critique that takes an input UI screenshot and design guidelines and generates a list of design comments, along with corresponding bounding boxes that map each comment to a specific region in the screenshot. The entire process is driven completely by VLMs, which iteratively refine both the text output and bounding boxes (in a mutually conditioned manner), using few-shot samples tailored for each step. We evaluated our approach using Gemini-1.5-pro and GPT-4o, and found that human experts generally preferred the design critiques generated by our pipeline over those by the baseline, with the pipeline reducing the gap from human performance by 50% for one rating metric. To assess its generalizability to other multimodal tasks, we applied our pipeline to open-vocabulary object and attribute detection, and experiments showed that our method also outperformed the baseline.

## 1 Introduction

Critiques are essential for design, providing feedback to help designers improve their work (Duan et al., 2024b; Wang et al., 2021; Xu et al., 2014). However, obtaining design critiques is often costly and time-consuming, hindering the design process. Hence, automating design critiques has become an important goal in many design fields. In this paper, we focus on automating critiques for user interface (UI) design—a prevalent task in industry that directly impacts the user experience (Stone et al., 2005). Obtaining UI design feedback typically requires expert reviews or user testing, which may be expensive and not always readily available. This makes automated critique very valuable, as it can provide instant feedback for designers to quickly iterate on (Duan et al., 2024b). Furthermore, automated design feedback can serve as a reward function for automated UI generation, which has started to gain traction (Zhao et al., 2021; Gajjar et al., 2021; Duan et al., 2024c).

UI design critique is often complex and open-ended, involving feedback that covers multiple dimensions of the design (e.g., aesthetics and usability) (Nielsen & Molich, 1990; Hartmann et al., 2008) and addresses both the overall design and specific problematic regions of the UI, based on design principles or guidelines. This makes automated UI critique a very challenging task. Given a UI screen and a set of design guidelines, the model needs to understand the screen, reason with UI design principles to detect violations in the UI design (both semantically and spatially), and explain and contextualize the feedback in a way that human designers can understand and act upon (Duan et al., 2024a) (Figure 1).

VLMs have made tremendous progress in a variety of multimodal tasks, such as visual question answering (VQA) and visual understanding, due to their extensive knowledge and generalization capabilities. Although VLMs appear to be readily usable for design critique, a multimodal task, there remains a significant gap in quality between the feedback generated by these VLMs compared to that of human design experts Duan et al. (2024a). In addition, VLMs often struggle to achieve

accurate visual grounding (Duan et al., 2024a; Dorkenwald et al., 2024), making it difficult for them to mark relevant regions in the UI screenshots, which is crucial for contextualizing feedback for designers (Duan et al., 2024a).

Recent advances in prompting techniques have improved both visual grounding and text generation performance. For example, Fang et al. (2024) introduced a visual prompting technique that adds visual markers to an image, which helps VLMs better ground objects. Separately, a method called *iterative refinement* Madaan et al. (2023); Xu et al. (2024a) has been proposed for text-only tasks, where an LLM's output is repeatedly refined by itself or another model until the output is deemed correct. *Iterative refinement* has been shown to improve performance for text-only tasks like code optimization and machine translation. Building on these, we propose a novel multimodal framework that combines both iterative refinement and visual prompting to generate UI design critiques (Figure 1). Our approach extends iterative refinement to multimodal tasks by jointly refining two coupled outputs: (1) the design critique text and (2) corresponding bounding boxes that ground the critique in the UI. Each refinement is conditioned on the other, i.e., text refinement is based on the grounding, and grounding refinement leverages the current text. This creates a feedback loop that promotes semantic and spatial alignment. To further improve grounding, we introduce visual prompting at several stages of the pipeline, including the iterative refinement steps. This enhances bounding box accuracy and, in turn, improves text accuracy. Our approach is implemented through an architecture that coordinates multiple VLMs (Figure 2) and incorporates both novel prompting techniques and established practices (Chen et al., 2024).

We evaluated our pipeline for UI critique using UICrit, a public dataset (Duan et al., 2024a), with two state-of-the-art VLMs: Gemini-1.5-pro (Team et al., 2024) and GPT-4o (OpenAI et al., 2024). Our experiments demonstrated that the pipeline consistently improved the design feedback output across both models, on both automatic metrics and human expert evaluation. To assess the broader applicability of our method to other multimodal tasks, we tested it on open-vocabulary object and attribute detection, where it consistently increased the mAP by up to 9.1. These experiments demonstrate the potential of our method to be useful in the broader scope of tasks, beyond design critique generation, pushing the boundary of what prompting can achieve for complex multimodal tasks.

## 2 RELATED WORK

### 2.1 AUTOMATED UI DESIGN CRITIQUE WITH LLMS AND VLMS

Prior work has studied the capabilities of LLMs/VLMs for UI design critique. Duan et al. (2024b) explored the performance of zero-shot (text-only) GPT-4 in critiquing UI mockups, using a JSON representation of the UI. They identified gaps between the feedback capabilities of general-purpose LLMs and human experts. To address this, they collected a dataset (UICrit) (Duan et al., 2024a) consisting of human-annotated design critiques (grounded within UI screenshots via bounding boxes) for UI screens that could be applied to train general-purpose VLMs. They also built a UI design critique model that takes in a UI screenshot and outputs critiques grounded in screenshot regions. Their method showed a significant improvement in VLM-generated feedback with just few-shot sampling from UICrit, although the feedback quality still falls short of human experts. Similarly, Wu et al. (2024) generated a synthetic dataset of UI design comments and trained a CLIP model (Radford et al., 2021) to assess UI designs. We apply our approach to the design critique task, which augments the method from Duan et al. (2024a) by incorporating mutually-conditioned iterative refinement of design comments and their corresponding bounding box positions on the UI screen.

### 2.2 PROMPTING LLMS WITH ITERATIVE REFINEMENT

Iterative refinement on LLM output has been explored in prior studies to improve LLM performance for text-only tasks. Madaan et al. (2023) developed an approach called "SELF-REFINE", where a single LLM generates an initial output and then iteratively provides feedback on its own output and revises the output based on the feedback. They applied this technique across a diverse set of tasks, such as math reasoning and dialogue response, and found that SELF-REFINE resulted in a 20% average performance gain. Similarly, Zhou et al. (2023) utilized this iterative self-refinement technique on long-horizon sequential task planning in robotics, leading to higher success rates. However, Xu et al. (2024b) found that LLMs often exhibit *self-bias* (i.e., a tendency to favor its own generated

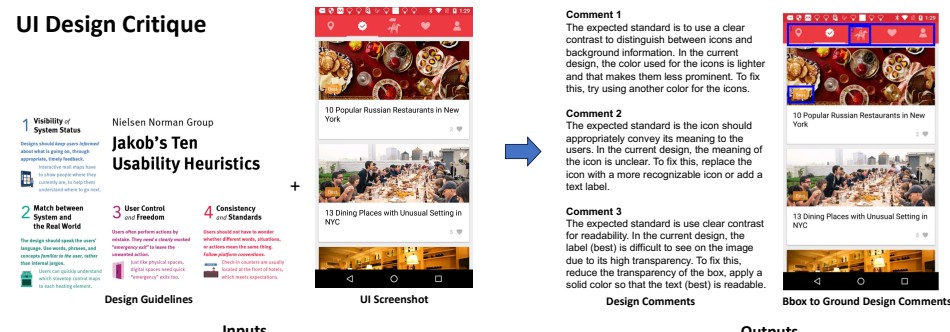

Figure 1: Illustration of the UI Design Critique Task, which takes in a UI screenshot and a set of design guidelines and outputs a list of design comments with corresponding bounding boxes (Bbox).

output) during self-refinement across a variety of tasks and languages. To account for this, they developed "LLMRefine" (Xu et al., 2024a), a method for text generation that uses a separate model to provide detailed feedback, along with a simulated annealing method to iteratively refine the LLM's output. We also utilize iterative refinement in our pipeline, and we extend this method to multimodal tasks by conditionally refining both text and bounding boxes that associate the text with relevant regions in the image. Following the method in LLMRefine, we use separate VLMs for generation and refinement to prevent self-bias.

## 2.3 MULTIMODAL TASKS

Previous work has investigated a variety of grounded multimodal tasks using VLMs, where a VLM takes in a visual input (such as an image) and generates outputs that are connected to specific objects, regions, or attributes within the visual input. Liu et al. (2023) introduced Grounding DINO, a transformer-based model that supports open-vocabulary object detection and can identify arbitrary objects within an image. Similarly, Bravo et al. (2023) introduced the Open Vocabulary Object and Attribute Detection task, which identifies and grounds both objects and their corresponding attributes in an image in an open vocabulary setting. In robotics, VLMs were used to help systems understand the physical world. Fang et al. (2024) introduced MOKA, which utilizes VLMs to solve complex robotic manipulation tasks by breaking them into multiple steps. Their approach incorporates visual prompting, where visual markers are added to the image, to aid in object grounding as part of the robot's step-by-step instructions. Chen et al. (2024) evaluated VLMs in the LLM-as-a-judge paradigm (Gu et al., 2025) across three tasks: pairwise comparison, scoring, and ranking. They found that while VLMs excelled at pairwise comparison, they struggled with the other tasks. However, Liu et al. (2024) showed that providing few-shot examples can improve an LLM's evaluation performance. Visual grounding is a vital component of our method, and we utilize visual prompting to enhance bounding box generation and refinement. We also utilize VLMs to validate output accuracy in our pipeline and use few-shot examples to enhance their evaluation accuracy.

## 3 TASK

*UI design critique generation* was introduced as a grounded multimodal task by Duan et al. (2024a). The model takes in a UI screenshot and a set of design guidelines and outputs a list of design critiques. Each design critique consists of a text comment that identifies a specific issue in the UI and a bounding box that highlights the relevant region of the screenshot (see Figure 1). For example, the text comment might state "*The expected standard is to use clear contrast for readability. In the current design, the label 'Best' is difficult to see on the image due to its high transparency. To fix this, reduce the transparency of the box and apply a solid color so that the text 'Best' is readable.*" and the bounding box will enclose the orange 'Best' tag in the UI screenshot in Figure 1.

As discussed earlier, this task is challenging because the model must understand and apply UI design principles to identify design issues in the screenshot. Furthermore, finding the exact region of the screen for a comment (i.e., the bounding box) is not always straightforward. For example, a com-

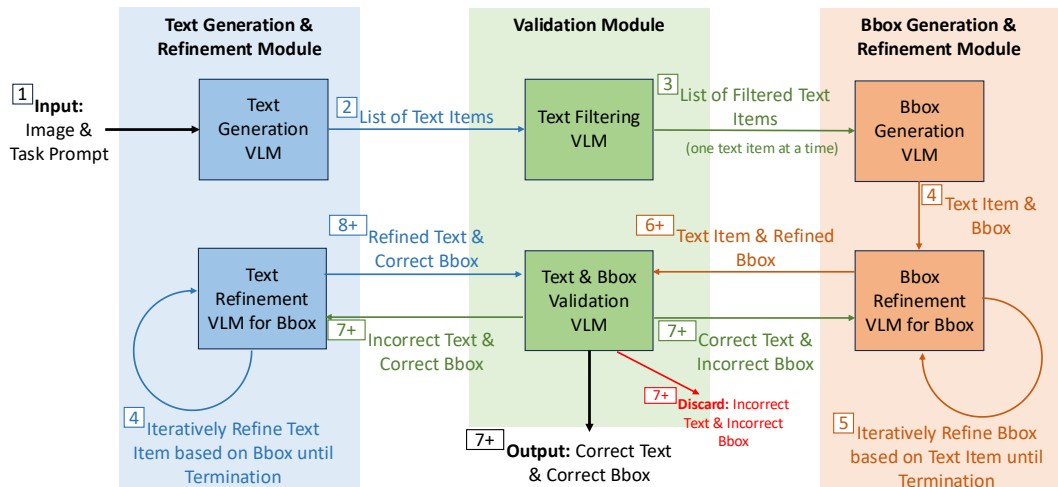

Figure 2: The figure illustrates our prompting pipeline, which takes an image and a task prompt as input and outputs text items with their corresponding bounding boxes on the image. The main inputs and outputs for each VLM are shown, and Section 4 details all the inputs, outputs, and few-shot examples for each VLM. Each input/output is numbered with their order of generation, and numbers with a '+' indicate multiple iterations of input/output.

ment might state that the text in the UI has poor contrast with the background, but not specify which text element is problematic, requiring the model to both identify the problematic elements and also determine their bounding box. While we focus on UI design critique, our task is representative of many multimodal tasks that require visually grounded text generation.

## 4 METHOD

We developed a prompting pipeline that uses multiple VLMs to generate UI design critiques. It consists of six distinct VLMs, organized into three modules: *Text Generation & Refinement*, *Validation*, and *Bounding Box Generation & Refinement*. These modules communicate with each other to complete the task. Figure 2 illustrates the pipeline, showing the main inputs and outputs of each VLM, which are numbered by the order of execution. We break down the entire task into separate generation and refinement steps for both text and bounding boxes, as decomposing complex tasks has been shown to improve performance (Khot et al., 2023).

As shown in the figure, each step's VLM output is conditioned on that of the previous step. Since Bounding Box Generation & Refinement is conditioned on the text predictions, and Text Refinement, in turn, is conditioned on the bounding box predictions, we introduce a Validation module (i.e., an LLM-as-a-judge) between the Text and Bounding Box modules to assess each module's output, ensuring each refinement step is based on more accurate inputs. Additionally, each VLM is provided with targeted few-shot examples to improve its accuracy, as well as a text prompt containing specific instructions for that step, which is derived from the input task prompt. To provide as much guidance as possible, we included the UI design guidelines in the input task prompt, which are also included in the instruction prompts for relevant steps. The specific inputs, outputs, and few-shot examples for each VLM are detailed in the following sections, and the instructions prompt for each step can be found in Appendix A.4. Appendix A.6 presents a cost analysis of the pipeline and visualizations of bounding box and comment refinements.

**Text Generation VLM (TextGen)** The pipeline begins with the TextGen VLM that takes an image and its instructions prompt (derived from the task prompt) as input, and generates a list of un-grounded text items (design comments) for the image. We decided to start with text generation and condition the bounding box generation on the generated text, instead of the other way around. This decision is based on our observation that for design critique, VLMs tend to perform poorly on visual grounding from scratch (i.e., without guidance from text), which makes the subsequent refinements much more error-prone.

**Text Filtering VLM (TextFilter)** To reduce the chance of bounding box generation being conditioned on incorrect text items (i.e., incorrect design comments), we utilize a separate VLM to assess each design comment's accuracy, operating in an LLM-as-a-judge role. This TextFilter VLM takes a list of generated text items from TextGen and the image as input, and outputs a filtered list of text items it considers valid. Based on the findings of Liu et al. (2024), we incorporate few-shot examples for TextFilter. These examples are constructed by injecting invalid items into a correct list of text items, and using this augmented list as input and the original correct list as the expected output. These examples illustrate how to filter out invalid items.

**Bounding Box Generation VLM (BoxGen)** The BoxGen VLM generates bounding boxes based on the filtered text items from TextFilter. The VLM takes in one text item at a time, as well as the image, and predicts a relevant region on the image via bounding box coordinates. Following the visual prompting technique from Duan et al. (2024a), we augment the screenshot by adding coordinate markers along its edges (Figure 4) to help the VLM associate coordinates with specific regions in the screen.

**Bounding Box Refinement VLM (BoxRefine)** To avoid self-bias during iterative refinement (Xu et al., 2024b), we use a separate VLM to iteratively refine the bounding box from the previous step, conditioned on the filtered text item. The BoxRefine VLM takes in several inputs, as shown in Figure 4 (Appendix). Similar to BoxGen, BoxRefine takes in the coordinate-marker enhanced screenshot image and a filtered text item. Additionally, BoxRefine takes in the bounding box coordinates that were predicted by BoxGen, and a close-up view of the image region specified by the predicted bounding box coordinates. In this zoomed-in image patch, the bounding box is displayed as a blue box, with some surrounding region of the box included for additional context. The zoomed-in image patch also has coordinate markers along the edges to help the LLM refine the bounding box coordinates based on this close-up view.

The VLM assesses the quality of the current bounding box based on all these inputs. If the bounding box is deemed accurate by the BoxRefine VLM, the iterative refinement process terminates. Otherwise, the VLM returns the refined coordinates, which are then re-evaluated by the VLM. This process is repeated until the VLM either confirms the bounding box as correct or the maximum number of iterations is reached. Prior work (Madaan et al., 2023) has shown that the history of refinements provides helpful information. Thus, we include the history of the VLM's refinements for the input bounding box as an input at each iteration, which enables the model to learn from past adjustments. Few-shot examples are generated by creating a synthetic refinement sequence with gradually reduced noise in the perturbation of a sampled bounding box's coordinates. Algorithm 1 (Appendix) details our methods for bounding box perturbation and the generation of few-shot examples for bounding box refinement.

**Text & Bounding Box Validation VLM (Validation)** After determining the bounding box for the text item, we again apply the LLM-as-a-judge framework through the Validation VLM, which determines if the bounding box and text are correct and can be used in the final output, or if they require further refinement. The Validation VLM takes as input the entire image, a zoomed-in image patch for the proposed region specified by the bounding box, and the text item, and assesses their accuracy. Figure 2 illustrates how the output is handled for each case: If both the text and bounding box are deemed correct, the pair is returned. If either the text or bounding box is incorrect, the pair is sent for further refinement. If both are incorrect, the pair is discarded, and the pipeline moves on to the next filtered text item. Few-shot examples are generated differently to handle each case; the bounding box is perturbed for the incorrect bounding box case (Algorithm 1 in the Appendix), the text item is perturbed for the incorrect text case, and both the bounding box and text are perturbed for the case where both are incorrect.

**Text Refinement VLM (TextRefine)** The TextRefine VLM is used to refine incorrect text items, conditioned on bounding boxes that correctly identify relevant regions in the image, as determined by the Validation VLM. This iterative refinement process mirrors the bounding box refinement procedure. The VLM takes as input the entire image, a zoomed-in image patch focused on the bounding box, and the text item, and refines the text iteratively until it determines that the text is accurate for the region shown in the bounding box. Few-shot examples are generated either by perturbing the text (if possible) or by selecting irrelevant text items from the few-shot dataset and then ranking them by increasing semantic similarity to the correct text, which simulates the refinement process. The refined text item and bounding box are then returned to the Validation VLM.

## 5 EXPERIMENTS

### 5.1 DATASET AND BASELINE

We used the UICrit dataset[1] (Duan et al., 2024a) to evaluate our pipeline for the design critique task. Each UI screenshot in this dataset was annotated by three experienced human designers, providing feedback that includes a list of text-based design critiques with their corresponding bounding boxes, numerical ratings for usability, aesthetics, and overall design quality, as well as a description of the screen's purpose. The dataset contains a total of 11,344 design critiques for 1,000 screenshots. For evaluation, we used the UI screenshots from UICrit as input images, included the three sets of design guidelines used by Duan et al. (2024a) in the task prompt, and evaluated the model's output against the comments and bounding boxes for the screenshot from the dataset (depending on the experiment). For few-shot examples, we sampled from a split of UICrit that is separate from the examples used for evaluation. The few-shot sampling method used at each step is detailed in A.3.1.

For the baseline, we used the few-shot pipeline developed by Duan et al. (2024a) for their UI critique task. Their pipeline consists of the Text Generation VLM (Figure 2) with few-shot sampling, followed by a VLM for bounding box generation that uses visual prompting (i.e., coordinates marked on the screenshot edges) without few-shot examples.

### 5.2 IMPACT OF VISUAL PROMPTING AND ITERATIVE REFINEMENT ON VISUAL GROUNDING

Table 1 presents an ablation study on the different components of the Bounding Box Generation and Refinement module (Figure 2), which illustrates the impact of visual prompting and iterative refinement on the visual grounding accuracy of two state-of-the-art VLMs: Gemini-1.5-pro and GPT-4o. For this evaluation, the module is given a UI screenshot and one of its comments from UICrit. Its output bounding box is evaluated against the ground-truth bounding box of that comment in UICrit by computing their IoU. The module consists of two VLMs (BoxGen and BoxRefine), and BoxRefine was only used for the conditions with iterative refinement.

For Gemini-1.5-pro, each enhancement led to an improvement in the average IoU, with the final setup (used in our pipeline) achieving an average IoU nearly three times higher than zero-shot and almost double that of zero-shot with visual prompting, which was used in the baseline (Duan et al., 2024a). For GPT-4o, improvements were seen at each step, except for zero-shot iterative refinement; when no few-shot examples were provided in the refinement prompt, GPT-4o did not refine any of the input bounding boxes. Additionally, while GPT-4o had better zero-shot performance, its IoU for the final setup was slightly worse than that of Gemini-1.5-pro. Nevertheless, iterative visual prompting led to substantial performance gains over zero-shot prompting for both VLMs, indicating that iterative visual prompting significantly enhances bounding box estimation.

### 5.3 PIPELINE ABLATION AND QUALITATIVE ANALYSIS

Table 2 presents the results of the ablation study for UI design critique for both VLMs, as well as the results for the baseline setup and multimodal Llama-3.2 11b (Dubey et al., 2024), which has been finetuned on the training split of UICrit for three epochs. Since UI design critique is open-ended, UICrit does not contain all the ground-truth design comments for each UI screenshot. Hence, we evaluated comment generation by computing the cosine similarity of sentenceBERT (Reimers & Gurevych, 2019) embeddings with each comment in the dataset for the UI screenshot and selecting the highest one ("Comment Similarity" in Table 2). The IoU was estimated by comparing the predicted bounding box with that of the most semantically similar comment ("Estimated IoU" in Table 2). The estimated IoU values are lower than those in Table 1, where the IoU was calculated directly from the input comments' corresponding bounding boxes in UICrit. The estimated IoU is lower because it uses the bounding box of the most semantically similar comment in the dataset instead, which may not precisely match the comment for which the bounding box was generated.

Each pipeline step incrementally improved comment similarity and estimated IoU for both VLMs. While GPT-4o and Gemini-1.5-pro showed similar comment similarity, GPT-4o achieved a higher estimated average IoU, likely due to its nearly three-fold larger parameter count. The complete

---

[1]https://github.com/google-research-datasets/uicrit

Table 1: IoU values from the Ablation study on the different components of bounding box generation. IR stands for Iterative Refinement, and VP stands for Visual Prompting.

| Methods | UI Critique IoU ↑ | |
|---|---|---|
| | Gemini$_{1.5tn}$ | GPT$_{4tn}$ |
| Zero-shot | 0.120 | 0.233 |
| Zero-shot, VP | 0.180 | 0.249 |
| Few-shot, VP | 0.267 | 0.319 |
| Few-shot, VP, Zero-shot IR | 0.279 | 0.319 |
| **Few-shot, VP, Few-shot IR** | **0.357** | **0.345** |

Table 2: Ablation study of our UI design critique pipeline. IR stands for Iterative Refinement. We combine the Validation step's results with subsequent refinements for bounding boxes and text, as these refinements apply to a much smaller subset of pairs flagged as having incorrect bounding boxes or text during Validation. We also include results from the Baseline and finetuned Llama-3.2 11b.

| Pipeline Steps | Comment Similarity ↑ | | Estimated IoU* ↑ | |
|---|---|---|---|---|
| | Gemini$_{1.5tn}$ | GPT$_{4tn}$ | Gemini$_{1.5tn}$ | GPT$_{4tn}$ |
| Text Generation | 0.651 | 0.680 | N/A | N/A |
| + Text Filtering | 0.694 | 0.692 | N/A | N/A |
| + Bbox Generation | 0.694 | 0.692 | 0.153 | 0.244 |
| + IR of Bbox | 0.694 | 0.692 | 0.173 | 0.259 |
| + Validation, IR of Text & Bbox | 0.702 | 0.701 | 0.199 | **0.275** |
| Baseline (Duan et al., 2024a) | 0.651 | 0.680 | 0.176 | 0.257 |
| Finetuned Llama-3.2 11b | **0.842** | | 0.230 | |

pipeline also outperforms the baseline in both comment similarity and estimated IoU. Fine-tuned Llama-3.2 achieves higher comment similarity than the pipeline, but its estimated IoU falls between those of Gemini-1.5-pro and GPT-4o for the complete pipeline.

While finetuned Llama-3.2 had higher comment similarity, it generated a very limited set of critiques, whereas our pipeline generated a considerably more diverse set of comments. To quantify diversity, we embedded design comments using sentenceBERT (Reimers & Gurevych, 2019) and computed the average distance (1 – cosine similarity) from each comment to the centroid, following the metric by Cox et al. (2021). The pipeline's diversity score (0.670) was considerably higher than finetuned Llama's (0.302). Example outputs and detailed qualitative analyses are provided in Appendix A.5.1. Appendix A.5.3 discusses simple ways to further refine the pipeline's output, such as using a third-party UI element detector (Xie et al., 2020) to clean up the bounding boxes.

## 5.4 HUMAN EVALUATION

Due to the open-ended nature of UI design critique, UICrit does not have the complete set of ground-truth design comments for each UI screen. Hence, we recruited human design experts to assess the validity of the feedback generated by our pipeline. For comparison, the experts also rated the comments generated by the *baseline* setup and human-annotated comments from UICrit. We used the same procedure devised by Duan et al. (2024a), where each design comment was rated as invalid, partially valid and valid, and the set of design comments from each condition was ranked as a whole, based on overall quality and comprehensiveness. Unlike the method used by Duan et al. (2024a), where participants rated both comment quality and bounding box accuracy together, our evaluation presented participants with a screenshot marked with a ground-truth bounding box (determined and agreed upon by the authors) and asked them to rate the validity of the comment only for that region. This ensures a more rigorous and standardized approach to evaluate bounding box accuracy and a more focused evaluation on comment quality. See Appendix A.7 for more details on the study method. Table 3 shows the average comment rating, the average comment set rank, and the average IoU for each of the three conditions for Gemini-1.5-pro's output. We used the established ground-truth bounding boxes from comments rated as valid or partially valid to compute the IoU with the predicted bounding boxes. For the "human" condition, the IoU was not computed as we displayed the bounding boxes from UICrit. The average Fleiss Kappa inter-rater reliability score (Fleiss et al.,

Table 3: Human expert ratings on UI design comments generated by Gemini-1.5-pro, and IoU of the generated bounding boxes for human validated comments.

| Methods | Comment Quality ↑ | Comment Set Rank ↓ | BBox IoU ↑ |
|---|---|---|---|
| Baseline (Duan et al., 2024a) | 0.45 | 2.3 | 0.423 |
| Our Pipeline | 0.47 | 2.0 | **0.451** |
| **Human** | **0.56** | **1.7** | N/A |

Table 4: Ablation study on the open vocabulary attribute detection (OVAD) and object detection (OVD) for Gemini-1.5-pro and GPT-4o. IR stands for Iterative Refinement. Note that bounding boxes are required for computing the mAP, so we combined the results for the text generation, text filtering, and bounding box generation steps. Similar to Table 2, we combined the results of the Validation step with additional iterative refinements of the bounding box and text.

| Pipeline Steps | OVAD mAP ↑ | | OVD mAP ↑ | |
|---|---|---|---|---|
| | $\text{Gemini}_{1.5tn}$ | $\text{GPT}_{4tn}$ | $\text{Gemini}_{1.5tn}$ | $\text{GPT}_{4tn}$ |
| Text Generation + Filtering + BBox | 11.3 | 13.1 | 13.1 | 15.8 |
| + IR of BBox | 12.6 | 14.0 | 15.8 | 17.8 |
| **+ Validation, IR of Comment & BBox** | **13.6** | **15.1** | **15.8** | **20.2** |
| Baseline | 11.1 | 12.9 | 11.2 | 11.1 |

1971) among the participants was 0.22 for comment quality and 0.29 for comment set ranking, indicating fair agreement.

Across all metrics, the pipeline outperformed the baseline, while human annotations remain the best. Interestingly, the average comment quality rating for human feedback was lower than expected, which may be attributed to the subjective nature of design critique (Nielsen & Molich, 1990) and variability in dataset quality, potentially due to UICrit's annotators' limited design experience (Duan et al., 2024a). While the gap between our pipeline and the baseline is modest, it still closes 22% of the gap between the baseline and human condition. Notably, the average comment set rank of our pipeline is positioned midway between the human and baseline setups, closing the gap from human performance by 50%. The comment set from our pipeline was preferred over the baseline's 58% of the time and was even favored over the human condition 38% of the time.

## 6 GENERALIZATION TO OTHER TASKS

Our pipeline can be applied to other multimodal tasks that output visually grounded text. To assess if its performance enhancement generalizes to other tasks, we evaluate our pipeline on an existing vision-language benchmark: Open Vocabulary Object and Attribute Detection Bravo et al. (2023).

### 6.1 OPEN VOCABULARY OBJECT & ATTRIBUTE DETECTION

Open vocabulary object and attribute detection Bravo et al. (2023) involves detecting objects and their associated attributes, along with bounding boxes marking their locations in the image (see Appendix A.1). During inference, the model is given a set of object classes and attributes to identify, including classes and attributes that were not seen during training (i.e., "open vocabulary"). Bravo et al. (2023) evaluated both attribute detection (OVAD) and object detection (OVD) in this open vocabulary setting. They collected a dataset[2] of human annotated object classes and attributes for 2,000 images from the MS COCO dataset Lin et al. (2014), including 80 object classes and 117 attribute categories. The object classes are divided into base and novel categories, with only the base classes seen during training. We used this dataset to evaluate our pipeline on this task. The task involves taking an image as input, along with a task prompt specifying the object and attribute classes. The output is evaluated against the ground truth object and attribute annotations. To meet the open-vocabulary criterion of this task, we sampled few-shot examples from the base classes only, from a split of their dataset, but used all the classes for evaluation. Appendix A.3.2 describes the fewshot sampling strategy in more detail.

---

[2] https://ovad-benchmark.github.io/

## 6.2 COMPARISON WITH BASELINE

Table 4 presents the results of the ablation study for open-vocabulary object and attribute detection, using both Gemini-1.5-pro and GPT-4o. We used the same baseline described in Section 5.1, as it can also be applied to this task. We followed the evaluation method from Bravo et al. (2023), calculating the mean average precision (mAP) across all attribute (OVAD) and object categories (OVD). The predicted text and corresponding bounding box were matched with the ground truth by selecting the bounding box with the highest IoU, with a minimum threshold of 0.5, and comparing the object categories and attribute classes.

Our approach outperformed the baseline mAP for OVAD by 2.5 and OVD by 4.6 with Gemini-1.5-pro, and by 2.2 for OVAD and 9.1 for OVD with GPT-4o. The larger performance gain for OVD may be due to the fact that it is a simpler task, with only 80 object categories compared to 117 attribute categories, and attributes are often more nuanced and harder to detect. Additionally, GPT-4o slightly outperformed Gemini-1.5-pro, likely due to its much larger size. However, our pipeline still falls short of the fine-tuned model from Bravo et al. (2023) (mAP 18.8 for OVAD and 39.3 for OVD).

# 7 DISCUSSION

Our pipeline outperforms the baseline for UI critique in both comment quality and grounding accuracy, based on automatic metrics (e.g., IoU) and human expert ratings; its feedback was also more often preferred by experts. This implies that the design feedback generated by our pipeline is more useful for human designers. Its performance improvement also generalizes to open-vocabulary object and attribute detection, suggesting the technique could potentially be applied to enhance other grounded multimodal tasks.

While our technique outperforms the baselines for open vocabulary object and attribute detection, it falls short of the fine-tuned LLMs from Bravo et al. (2023). This is expected, since our pipeline does not involve parameter-tuning, whereas their fine-tuned LLMs were trained on significantly more data than the few-shot examples provided to our model. For design critique, our pipeline generates a significantly more diverse set of critiques compared to finetuned Llama 3.2, potentially making our pipeline more useful in practice. However, our pipeline still has room for improvement when compared to human experts. Despite its performance gap with human critique (which is expensive to acquire), our pipeline's generalizability and consistent improvements over baselines, achieved solely through prompting without parameter tuning nor third-party models, demonstrate its potential as a versatile and resource-efficient solution for enhancing VLM performance across various tasks and domains.

A reason for the performance gap could be that the VLM-based validation steps are not fully accurate (Shankar et al., 2024; Chen et al., 2024), which could lead to incorrect judgement of the bounding box and/or text accuracy. Future work can improve the validation step with better prompting strategies, or look into a human-in-the-loop approach, where human experts validate or refine the text and bounding boxes. This human-in-the-loop validation could both improve the immediate quality of the output and help the system learn from human inputs over time, via targeted few-shot examples. This step can be integrated into a design tool where designers validate or refine the feedback, so the model learns to provide more accurate and personalized design critiques over time.

# 8 CONCLUSION

We introduce a novel prompting pipeline that improves both the quality and visual grounding of automated UI design critique through visual prompting and conditional iterative refinement of both text and bounding boxes. Our approach outperformed the baseline in human evaluations, generating higher quality comments with more accurate grounding. We also demonstrated the generalizability of our technique through performance gains in open-vocabulary object and attribute detection, suggesting its potential to enhance other grounded multimodal tasks. Despite its limitations, our method offers a versatile and resource-efficient solution for improving VLM performance across various tasks and domains.

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

# A APPENDIX

## A.1 OPEN VOCABULARY OBJECT AND ATTRIBUTE DETECTION TASK

Open vocabulary object and attribute detection, developed by Bravo et al. (2023), is a benchmark task that involves detecting objects and their associated attributes, along with bounding boxes marking their locations in the image. Figure 3 shows an example for the Open Vocabulary Object and Attribute Detection Task. For further details about the task and the dataset, see the original paper (Bravo et al., 2023).

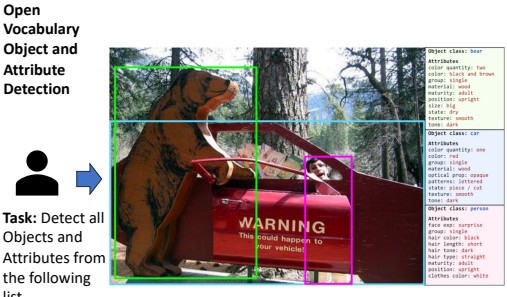

Figure 3: Illustration of the Open Vocabulary Object and Attribute Detection Task. The example output is taken from Bravo et al. (2023).

## A.2 ILLUSTRATION OF INPUTS FOR BOUNDING BOX REFINEMENT

Figure 4 shows an example set of inputs for a Bounding Box Refinement VLM (BoxRefine) call.

## A.3 FEW-SHOT SAMPLING METHODS FOR BOTH TASKS

### A.3.1 UI DESIGN CRITIQUE

For both design comment generation and filtering, we sampled UI screenshots and corresponding comments based on UI task and visual similarity from a split of UICrit, following the best-performing sampling method from Duan et al. (2024a). We used CLIP (Radford et al., 2021) to generate joint task and screenshot embeddings, and cosine similarity to determine relatedness. For filtering, we augmented the dataset's comments with VLM-generated comments deemed incorrect by annotators (Duan et al. (2024a)). For bounding box generation, refinement, and subsequent steps that operate on individual comments, we sampled few-shot examples by selecting the most semantically similar comments and their corresponding bounding boxes from a split of UICrit. We used sentenceBERT (Reimers & Gurevych, 2019) to embed the comment text for similarity ranking. For validation, few-shot examples of invalid comments were selected from incorrect comments that were marked by dataset annotators, or from irrelevant comments from other UIs. Finally, for text refine-

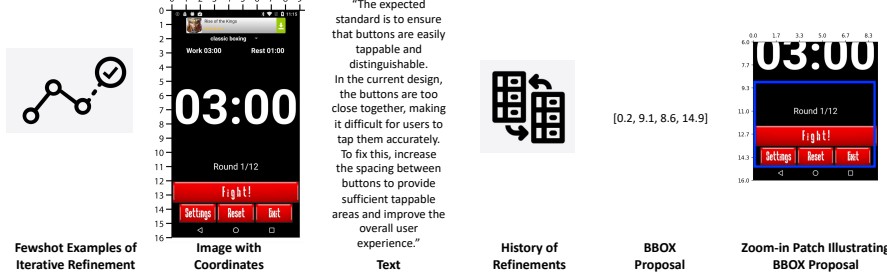

Figure 4: An example of the inputs to the Bounding Box Refinement VLM.

ment, multiple invalid comments were selected, following the process described earlier, and then sorted by increasing cosine similarity to simulate the comment refinement process.

For bounding box refinement, we considered another technique to generate fewshot examples. This technique involves selecting the first bounding box location based on visual similarity of the region it contains in the fewshot UI to that of the region contained by the input bounding box proposal of the input screenshot. This bounding box is then gradually moved closer to the ground truth bounding box for the fewshot UI to simulate the refinement process. However, we found that the simpler approach of randomly perturbing the bounding box actually gave better results (IoU 0.357 (random perturbation, from Table 1) vs 0.333 (visual similarity match)).

### A.3.2 Open Vocabulary Object and Attribute Detection

For text generation (i.e., category and attributes) and filtering, we sampled images based on the semantic similarity of their CLIP embeddings. Negative text samples for the filtering step were generated by sampling irrelevant text from other images. For bounding box generation, refinement, and subsequent steps applied to individual text items, we sampled few-shot examples by selecting the most semantically similar text items and their corresponding bounding boxes from a split of their annotated dataset. We used sentenceBERT (Reimers & Gurevych, 2019) to embed the text items for similarity ranking. For validation, invalid text examples were perturbed by randomly swapping the category or attributes, or by deleting or adding attributes. Similarly, for text refinement, few-shot examples were generated by perturbing the text in decreasing amounts.

### A.4 Instructions Prompts for Pipeline

We provide the instructions prompt for each step of the pipeline for the UI Critique Task.

```
Text Generation: For these sets of guidelines: [Guidelines]. Please find
    all the guideline violations in the UI provided. For violation found,
    please provide an explanation that includes these three things: 1.
    the expected standard (i.e. what good design should look like), 2.
    the gap between the current design and the expected standard (i.e.
    the critique for the design), and 3. how to fix the issue in the
    current design. For formatting each violation, please include these
    three things in separate sentences. For the expected standard (#1),
    start the sentence with 'The expected standard is that...'. For the
    gap (#2), start the sentence with 'In the current design, ...', and
    for how to fix the design (#3), start the sentence with 'To fix this
    ...'. Please end each violation explanation with two newline
    characters (\n\n). Please be specific in your violation explanations,
    making sure to refer to specific UI elements and groups in the UI.
    After determining all guideline violations, please also share any
    other design feedback you have for the UI and follow the same format
    of providing the expected standard, the critique for the design, and
    how to fix the issue. We will provide N examples of a UI screenshot
    and a set of valid design comments. Please learn how to give valid
    design comments from these examples and apply this knowledge to
    determine valid design comments for the last UI. Please be specific
    in your comments, referring to specific UI elements by their text
    label or icon, like in the examples provided. Also, please do not
    return any comments regarding user testing nor adherence to platform
    standards.

Text Filtering: For the provided UI and a list corresponding design
    comments, please filter out the incorrect design comments and return
    a list tuples. Each tuple contains its index i in the list, followed
    by True or False. The tuple would contain True if the design comment
    at index i in the input list is a valid design comment, and False if
    the design comment at index i is an invalid comment. Please analyze
    the UI screenshot to determine whether or not each design comment is
    valid. We will give N examples, where each UI screenshot is followed
    by a list of its corresponding design comments and an output list of
    tuples, where each tuple contains the list index and True/False
```

indicating the validity of the design comment at that index. Please
learn from these examples, analyzing the UI screenshot to see why
each comment was considered valid or invalid. Finally, we will give a
UI screenshot, followed by its corresponding design comments. Please
output a list of tuples consisting of the comment's list index and
an indication of each comment's validity, like in the provided
examples. Please output False for the design comment if it is about
consistency with the brand, user testing, or adherence to platform
standards. Please only output this list of tuples and nothing else.

Bounding Box Generation: You will be providing bounding boxes coordinates
for the provided UI screenshot and design comment. The bounding box
will enclose a relevant region in the screenshot that is discussed in
the design comment. You will use the coordinate axes along the edge
of the screenshot to determine the coordinates of the bounding box.
Please make sure you follow the provide coordinate axes, so that
vertical bounding box coordinates are between 0 and 16 and horizontal
bounding box coordinates are between 0 and 9, and format the
bounding box coordinates as (left, top, right, bottom). Please do not
output bounding boxes with area 0. Also, please only output the
bounding box and nothing else. We will provide N examples of design
comments, followed by the corresponding UI screenshot (with a
coordinate axis along its edge) and a correct bounding box for the
design comment in the UI screenshot based on the coordinate axis.
Please learn how to determine accurate bounding boxes for the design
comment in the UI screenshot based on these examples. We will provide
a final design comment and UI screenshot; please apply what you have
learned from the examples to determine an accurate bounding box for
this final design comment and UI screenshot only.

Bounding Box Refinement: You will be refining bounding boxes for a given
UI screenshot and design comment. The bounding box will enclose a
relevant region in the screenshot that is discussed in the design
comment. You will be given a proposed bounding box candidate and will
evaluate whether or not this bounding box accurately encloses the
region in the screenshot that is discussed in the comment. The
proposed bounding box coordinates, in the format of (left_coordinate,
top_coordinate, right_coordinate, bottom_coordinate) and is
displayed as a blue box in the screenshot patch that is also provided
, with some additional margin around the blue bounding box. Please
reflect on whether or not this bounding box is accurate and look
closely at the UI elements contained in the blue bounding box to
judge its accuracy and relevance to the design comment. If the
bounding box is not accurate, please output a new bounding box that
you think is accurate in the format of (left_coordinate,
top_coordinate, right_coordinate, bottom_coordinate), where each
coordinate is determined from the coordinate axes along the edge of
the UI screenshot provided earlier. Please make sure the new bounding
box you output is accurate, and refer to the coordinate axes along
the edge of the zoomed-in screenshot patch and the entire screenshot
(provided earlier) to determine the bounding box coordinates. If the
bounding box is accurate, please output 'BOUNDING BOX IS ACCURATE,
PLEASE TERMINATE'. Please only output either the updated bounding or
'BOUNDING BOX IS ACCURATE, PLEASE TERMINATE' and nothing else. We
will provide N examples of bounding box refinements for a given
design comment, UI screenshot, and bounding box candidate. Please
learn how to accurately refine bounding boxes for the design comment
in the UI screenshot based on these examples. We will provide a final
design comment, UI screenshot, and bounding box candidate; please
apply what you have learned from the examples to accurately refine
the bounding box candidate for this final design comment, UI
screenshot, and the zoomed in patch showing the bounding box
candidate.

Text and Bounding Box Validation: You are given a UI screenshot, design comment for the UI screen, and a zoomed-in patch of the UI screenshot showing the corresponding bounding box for the design comment. Please evaluate the accuracy of the design comment and bounding box with respect to the UI screenshot. The bounding box is displayed as a blue box in the zoomed-in screenshot patch, and is supposed to contain the region in the UI screen that is targeted by the design comment. Please first evaluate if the design comment is valid for the provided UI screenshot, i.e. if it correctly points out a design issue and suggests an accurate way to fix it. Please analyze the provided UI screenshot to assess the comment's validity. If the design comment is valid, please next evaluate whether the blue box in zoomed-in UI screenshot contains the region that is relevant to the design comment. If the design comment is invalid and the blue box still contains a region in the UI screenshot with design issues, please return the label 'Incorrect Comment'. If the comment is valid, but the blue box does not contain the region relevant to the comment, please return the label 'Incorrect Bbox'. If the comment is invalid and the blue box does not contain a region with design issues, please return the label 'Both Incorrect'. Finally, if the design comment is valid and the blue box correctly contains a region in the UI that is relevant to the comment, please return the label 'Both Correct'. Please only return the appropiate label and nothing else. We will give N examples, the UI screenshot (labeled 'UI Screenshot'), followed by the design comment (labeled 'Design Comment'), a zoomed-in screenshot patch showing the blue bounding box (labeled 'Zoomed-in Patch'), and finally the correct label (labeled 'Label') for the accuracy of the UI screenshot, design comment, and corresponding bounding box. Please learn from these examples, to see how to correctly categorize the design comment and its corresponding bounding box by accuracy. Finally, we will give a UI screenshot, design comment, and a zoomed-in patch showing the corresponding blue bounding box. Please apply what you have learned from the examples to correctly classify the accuracy of the design comment and its corresponding bounding box.

Text Refinement: You will be refining the design comment for a specific region in a UI screenshot. You will be given a UI screenshot, a zoomed-in patch of the screenshot with a blue box containing the region of interest, and a design comment for the UI region inside the blue box. Please evaluate whether or not the design comment accurately describes the design issue for the UI region inside the blue box. If the design comment is accurate, please output 'COMMENT IS ACCURATE, PLEASE TERMINATE'. If the design comment is not accurate, please refine the design comment to the accurate and output this accurate design comment for the region of interest, following the same format as the input design comment. We will provide N examples of bounding box refinements for each UI screenshot, region of interest, and design comment candidate for the region of interest. Please learn how to accurately refine the design comment for the region of interest in the UI screenshot based on these examples. We will provide a final UI screenshot, region of interest, and design comment candidate for the region of interest; please apply what you have learned from the examples to accurately refine design comment candidate for this final UI screenshot and region of interest. Please only output the refined comment or 'COMMENT IS ACCURATE, PLEASE TERMINATE' and nothing else.

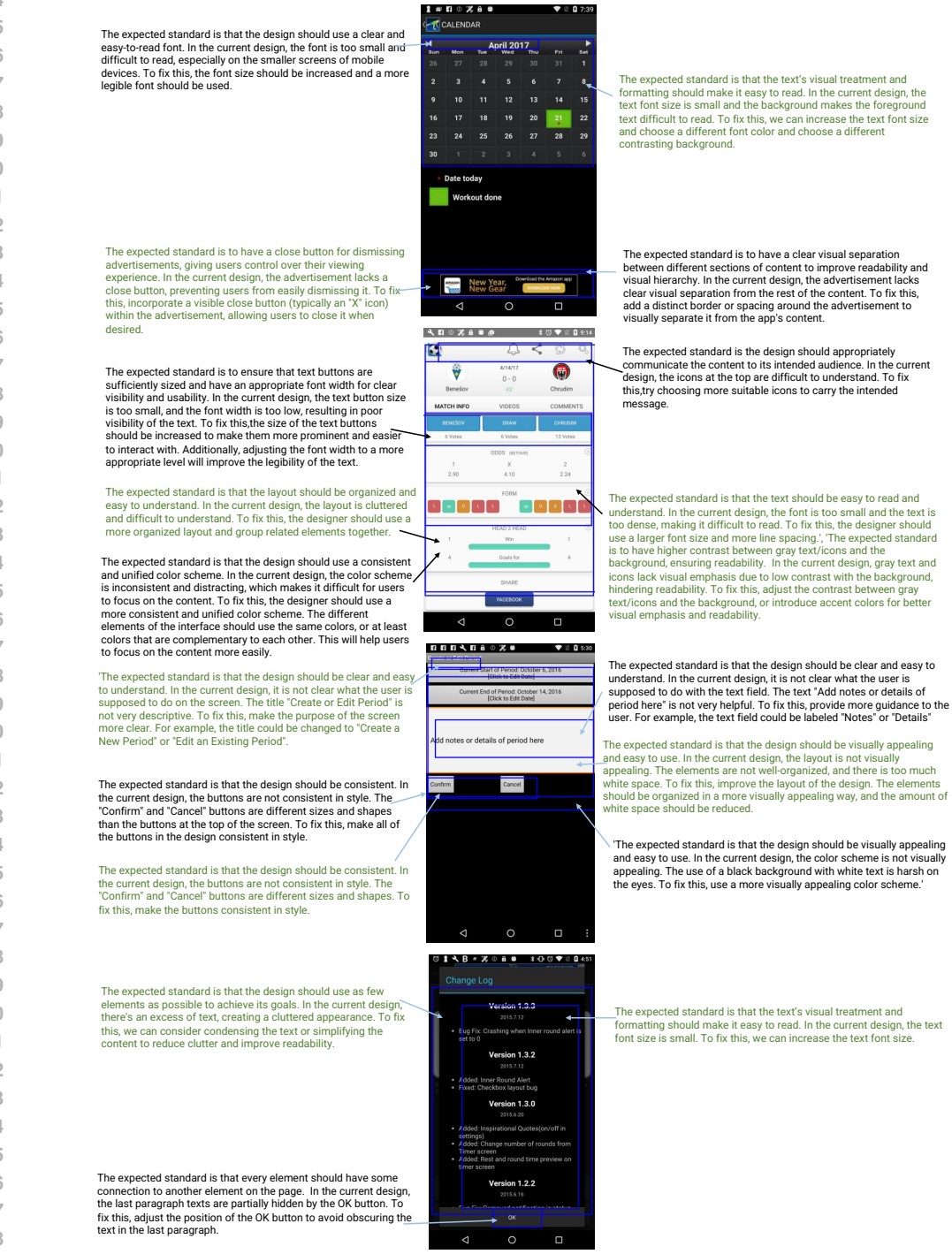

Figure 5: Illustration of four example outputs from the pipeline. The screenshots are marked with the output bounding boxes, and the generated comments are shown, each pointing to its corresponding bounding box. Helpful comments with reasonably accurate bounding boxes are highlighted in screen.

## A.5 Qualitative Analysis

### A.5.1 Qualitative Analysis of Outputs from Pipeline, Baseline, and Finetuned VLM

We qualitatively analyzed the outputs from our pipeline, baseline, and finetuned Llama-3.2 11b. Figures 5, 6, and 7 illustrate the design feedback and corresponding bounding boxes generated by our

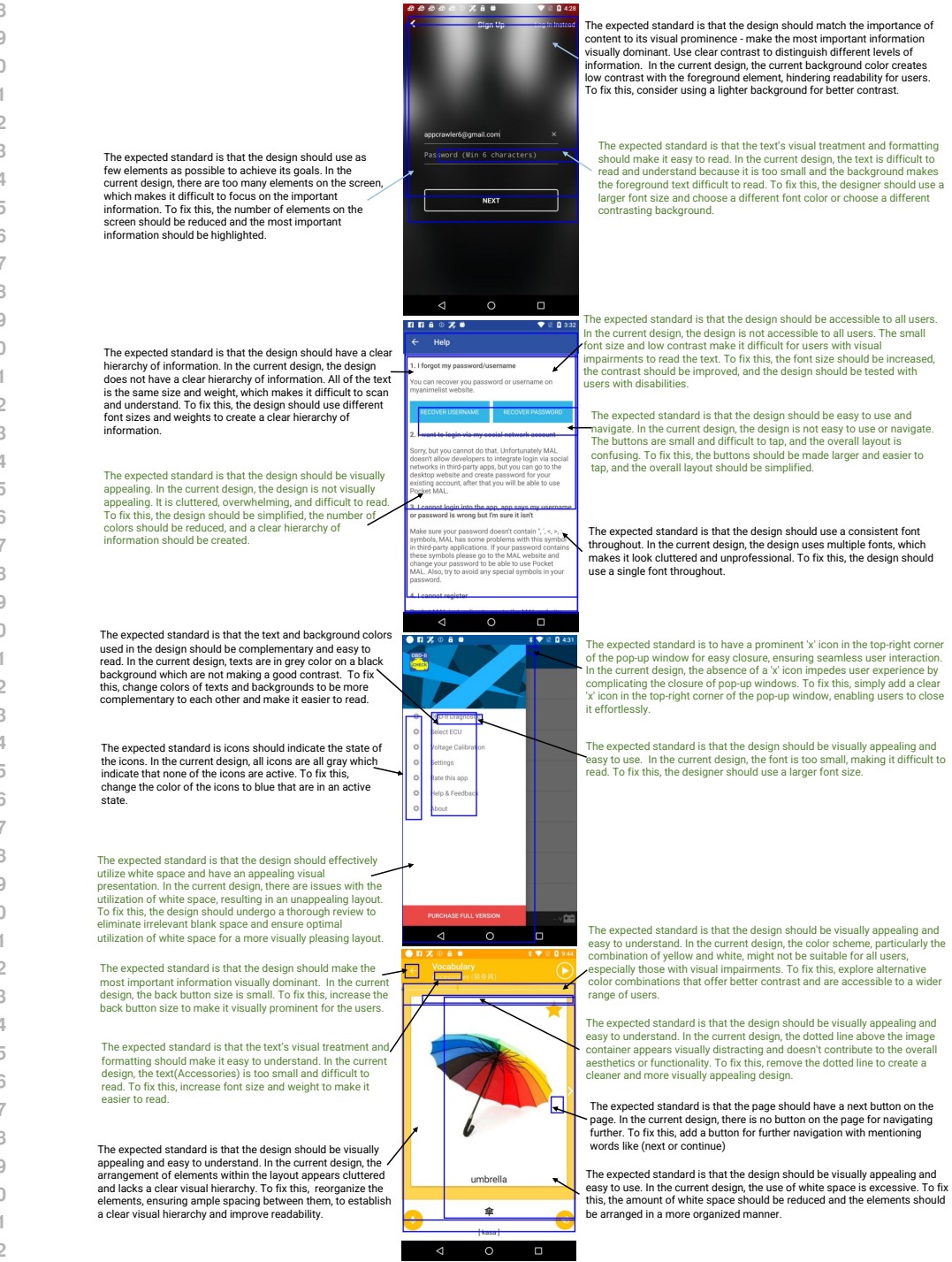

The expected standard is that the design should use as few elements as possible to achieve its goals. In the current design, there are too many elements on the screen, which makes it difficult to focus on the important information. To fix this, the number of elements on the screen should be reduced and the most important information should be highlighted.

The expected standard is that the design should match the importance of content to its visual prominence - make the most important information visually dominant. Use clear contrast to distinguish different levels of information. In the current design, the current background color creates low contrast with the foreground element, hindering readability for users. To fix this, consider using a lighter background for better contrast.

The expected standard is that the text's visual treatment and formatting should make it easy to read. In the current design, the text is difficult to read and understand because it is too small and the background makes the foreground text difficult to read. To fix this, the designer should use a larger font size and choose a different font color or choose a different contrasting background.

The expected standard is that the design should have a clear hierarchy of information. In the current design, the design does not have a clear hierarchy of information. All of the text is the same size and weight, which makes it difficult to scan and understand. To fix this, the design should use different font sizes and weights to create a clear hierarchy of information.

The expected standard is that the design should be accessible to all users. In the current design, the design is not accessible to all users. The small font size and low contrast make it difficult for users with visual impairments to read the text. To fix this, the font size should be increased, the contrast should be improved, and the design should be tested with users with disabilities.

The expected standard is that the design should be easy to use and navigate. In the current design, the design is not easy to use or navigate. The buttons are small and difficult to tap, and the overall layout is confusing. To fix this, the buttons should be made larger and easier to tap, and the overall layout should be simplified.

The expected standard is that the design should be visually appealing. In the current design, the design is not visually appealing. It is cluttered, overwhelming, and difficult to read. To fix this, the design should be simplified, the number of colors should be reduced, and a clear hierarchy of information should be created.

The expected standard is that the design should use a consistent font throughout. In the current design, the design uses multiple fonts, which makes it look cluttered and unprofessional. To fix this, the design should use a single font throughout.

The expected standard is that the text and background colors used in the design should be complementary and easy to read. In the current design, texts are in grey color on a black background which are not making a good contrast. To fix this, change colors of texts and backgrounds to be more complementary to each other and make it easier to read.

The expected standard is to have a prominent 'x' icon in the top-right corner of the pop-up window for easy closure, ensuring seamless user interaction. In the current design, the absence of a 'x' icon impedes user experience by complicating the closure of pop-up windows. To fix this, simply add a clear 'x' icon in the top-right corner of the pop-up window, enabling users to close it effortlessly.

The expected standard is icons should indicate the state of the icons. In the current design, all icons are gray which indicate that none of the icons are active. To fix this, change the color of the icons to blue that are in an active state.

The expected standard is that the design should be visually appealing and easy to use. In the current design, the font is too small, making it difficult to read. To fix this, the designer should use a larger font size.

The expected standard is that the design should effectively utilize white space and have an appealing visual presentation. In the current design, there are issues with the utilization of white space, resulting in an unappealing layout. To fix this, the design should undergo a thorough review to eliminate irrelevant blank space and ensure optimal utilization of white space for a more visually pleasing layout.

The expected standard is that the design should make the most important information visually dominant. In the current design, the back button size is small. To fix this, increase the back button size to make it visually prominent for the users.

The expected standard is that the design should be visually appealing and easy to understand. In the current design, the color scheme, particularly the combination of yellow and white, might not be suitable for all users, especially those with visual impairments. To fix this, explore alternative color combinations that offer better contrast and are accessible to a wider range of users.

The expected standard is that the text's visual treatment and formatting should make it easy to understand. In the current design, the text(Accessories) is too small and difficult to read. To fix this, increase font size and weight to make it easier to read.

The expected standard is that the design should be visually appealing and easy to understand. In the current design, the dotted line above the image container appears visually distracting and doesn't contribute to the overall aesthetics or functionality. To fix this, remove the dotted line to create a cleaner and more visually appealing design.

The expected standard is that the page should have a next button on the page. In the current design, there is no button on the page for navigating further. To fix this, add a button for further navigation with mentioning words like (next or continue)

The expected standard is that the design should be visually appealing and easy to understand. In the current design, the arrangement of elements within the layout appears cluttered and lacks a clear visual hierarchy. To fix this, reorganize the elements, ensuring ample spacing between them, to establish a clear visual hierarchy and improve readability.

The expected standard is that the design should be visually appealing and easy to use. In the current design, the use of white space is excessive. To fix this, the amount of white space should be reduced and the elements should be arranged in a more organized manner.

Figure 6: Illustration of four example outputs from the pipeline. The screenshots are marked with the output bounding boxes, and the generated comments are shown, each pointing to its corresponding bounding box. Helpful comments with reasonably accurate bounding boxes are highlighted in screen.

pipeline (using Gemini-1.5-pro) for a diverse set of 12 UIs. Figure 8 presents two examples where our pipeline outperformed the baseline, and Figure 9 contains two examples where the baseline performed better. To enable easier comparison between the two conditions, we used the same set

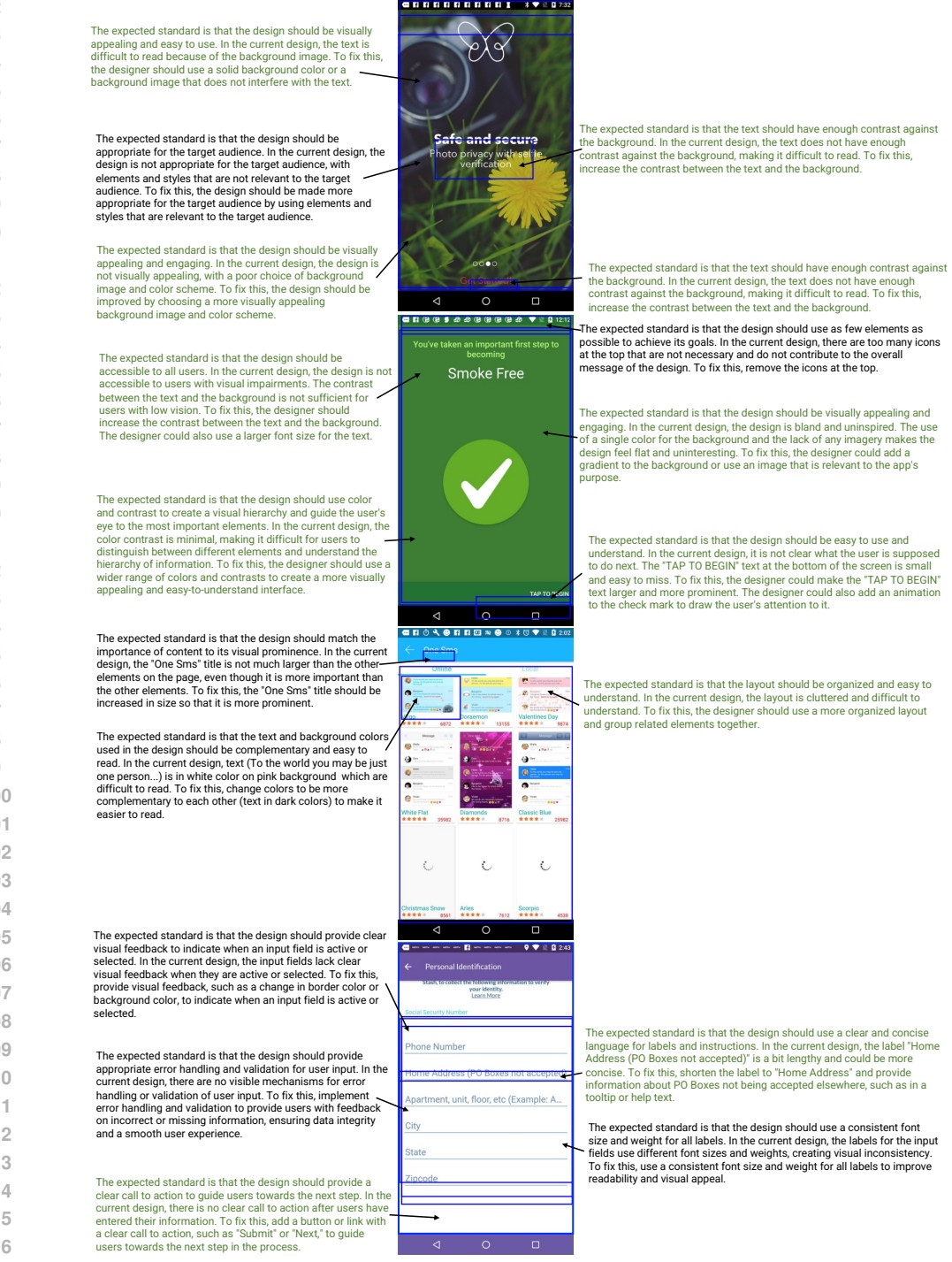

The expected standard is that the design should be visually appealing and easy to use. In the current design, the text is difficult to read because of the background image. To fix this, the designer should use a solid background color or a background image that does not interfere with the text.

The expected standard is that the design should be appropriate for the target audience. In the current design, the design is not appropriate for the target audience, with elements and styles that are not relevant to the target audience. To fix this, the design should be made more appropriate for the target audience by using elements and styles that are relevant to the target audience.

The expected standard is that the design should be visually appealing and engaging. In the current design, the design is not visually appealing, with a poor choice of background image and color scheme. To fix this, the design should be improved by choosing a more visually appealing background image and color scheme.

The expected standard is that the design should be accessible to all users. In the current design, the design is not accessible to users with visual impairments. The contrast between the text and the background is not sufficient for users with low vision. To fix this, the designer should increase the contrast between the text and the background. The designer could also use a larger font size for the text.

The expected standard is that the design should use color and contrast to create a visual hierarchy and guide the user's eye to the most important elements. In the current design, the color contrast is minimal, making it difficult for users to distinguish between different elements and understand the hierarchy of information. To fix this, the designer should use a wider range of colors and contrasts to create a more visually appealing and easy-to-understand interface.

The expected standard is that the design should match the importance of content to its visual prominence. In the current design, the "One Sms" title is not much larger than the other elements on the page, even though it is more important than the other elements. To fix this, the "One Sms" title should be increased in size so that it is more prominent.

The expected standard is that the text and background colors used in the design should be complementary and easy to read. In the current design, text (To the world you may be just one person...) is in white color on pink background which are difficult to read. To fix this, change colors to be more complementary to each other (text in dark colors) to make it easier to read.

The expected standard is that the design should provide clear visual feedback to indicate when an input field is active or selected. In the current design, the input fields lack clear visual feedback when they are active or selected. To fix this, provide visual feedback, such as a change in border color or background color, to indicate when an input field is active or selected.

The expected standard is that the design should provide appropriate error handling and validation for user input. In the current design, there are no visible mechanisms for error handling or validation of user input. To fix this, implement error handling and validation to provide users with feedback on incorrect or missing information, ensuring data integrity and a smooth user experience.

The expected standard is that the design should provide a clear call to action to guide users towards the next step. In the current design, there is no clear call to action after users have entered their information. To fix this, add a button or link with a clear call to action, such as "Submit" or "Next," to guide users towards the next step in the process.

The expected standard is that the text should have enough contrast against the background. In the current design, the text does not have enough contrast against the background, making it difficult to read. To fix this, increase the contrast between the text and the background.

The expected standard is that the text should have enough contrast against the background. In the current design, the text does not have enough contrast against the background, making it difficult to read. To fix this, increase the contrast between the text and the background.

The expected standard is that the design should use as few elements as possible to achieve its goals. In the current design, there are too many icons at the top that are not necessary and do not contribute to the overall message of the design. To fix this, remove the icons at the top.

The expected standard is that the design should be visually appealing and engaging. In the current design, the design is bland and uninspired. The use of a single color for the background and the lack of any imagery makes the design feel flat and uninteresting. To fix this, the designer could add a gradient to the background or use an image that is relevant to the app's purpose.

The expected standard is that the design should be easy to use and understand. In the current design, it is not clear what the user is supposed to do next. The "TAP TO BEGIN" text at the bottom of the screen is small and easy to miss. To fix this, the designer could make the "TAP TO BEGIN" text larger and more prominent. The designer could also add an animation to the check mark to draw the user's attention to it.

The expected standard is that the layout should be organized and easy to understand. In the current design, the layout is cluttered and difficult to understand. To fix this, the designer should use a more organized layout and group related elements together.

The expected standard is that the design should use a clear and concise language for labels and instructions. In the current design, the label "Home Address (PO Boxes not accepted)" is a bit lengthy and could be more concise. To fix this, shorten the label to "Home Address" and provide information about PO Boxes not being accepted elsewhere, such as in a tooltip or help text.

The expected standard is that the design should use a consistent font size and weight for all labels. In the current design, the labels for the input fields use different font sizes and weights, creating visual inconsistency. To fix this, use a consistent font size and weight for all labels to improve readability and visual appeal.

Figure 7: Illustration of four example outputs from the pipeline. The screenshots are marked with the output bounding boxes, and the generated comments are shown, each pointing to its corresponding bounding box. Helpful comments with reasonably accurate bounding boxes are highlighted in screen.

of initial comments from the TextGen module, as both the pipeline and baseline begin with this module.

As shown in figures 5, 6, and 7, we found that, more often than not, the pipeline generates helpful comments with reasonably accurate bounding boxes (highlighted in green). For the baseline, we observed that it frequently generates very generic comments that would apply to any UI screen and are usually not helpful, such as suggesting that at design should be tested with users or needs to be made responsive as shown in Figure 8 (Baseline, top screenshot). These comments are usually eliminated by the pipeline (Pipeline, top screenshot). Additionally, the pipeline successfully refined incorrect comments, as shown by the red and green comments in the top screenshot, and filters out incorrect comments during the validation stages as shown in both screenshots. For bounding boxes, those generated by the pipeline are usually tighter and closer to the correct region compared to the baseline, which often generates large, unspecific bounding boxes that encompass a significant portion of the screen, as shown by the bounding boxes in Figures 8 and 9. This demonstrates the effectiveness of iterative refinement and validation in improving bounding box accuracy. Furthermore, the large bounding boxes generated by the baseline would decrease the chance of the IoU being zero, which may have inflated the average IoU shown in Tables 2 and 3.

The pipeline sometimes eliminated valid comments, as shown in both examples in Figure 9, where the green comments were accurate comments that were eliminated. In the top screenshot, the pipeline retained only one inaccurate comment, although its bounding box was significantly improved. In the bottom screenshot, the pipeline produced a less accurate bounding box around the red buttons compared to the baseline, though these instances are rare.

We found that fine-tuned Llama-3.2 generated a very limited range of comments, primarily focusing on text readability, visual clutter, and generic critiques about the need for improved visual appeal. This limited range could be due to the over-representation of such critiques in UICrit. Figure 10 presents example outputs for two screenshots, comparing them with outputs from our pipeline. The figure shows that, in addition to its limited range of critiques, the finetuned model also produces inaccurate comments. In contrast, our pipeline generates a significantly more diverse set of comments with tighter bounding boxes, though the bounding boxes are generally less accurate than those from the fine-tuned model.

Overall, the pipeline generally outperforms the baseline qualitatively, reducing the generation of invalid and generic comments and outputting bounding boxes that are tighter, more specific, and closer to the target region. Furthermore, it generates a considerably more diverse set of comments compared to finetuned Llama, though its visual grounding is less accurate.

### A.5.2 Qualitative Analysis of Pipeline Outputs for Out of Domain UIs

Since UICrit consists of older UIs (from 2014) Duan et al. (2024a), we evaluated the pipeline's performance to determine whether it generalizes to modern UIs and other out-of-domain UIs, such as websites, using only few-shot examples selected from UICrit. Figure 11 displays the generated feedback for four modern Android UIs (the few-shot examples from UICrit are also Android UIs) from 2024, taken from Mobbin[3]. Figure 12 presents feedback for four modern iOS UIs from 2024, sourced from DesignVault[4], and Figure 13 illustrates feedback for five modern websites from 2024, also taken from Mobbin. In these figures, helpful comments with reasonably accurate bounding boxes are highlighted in green.

The pipeline was able to provide helpful feedback with reasonably accurate bounding boxes for these out-of-domain UIs. It performed surprisingly well on the modern iOS UIs, with results comparable to those for the UIs from the test split of UICrit, as shown in Figures 5, 6, 7, and 8. Additionally, the pipeline even managed to generate helpful feedback and bounding boxes for websites. While design principles often overlap between mobile and web interfaces, their layouts and screenshot dimensions differ significantly. This suggests that the VLM was able to generalize and adapt its knowledge to generate and refine bounding boxes for website screenshots, despite only being trained with few-shot examples from mobile screenshots, which are very different.

An interesting observation is that, since websites have more screen space, they are generally more complex and information-dense than mobile UIs (Gazzawe, 2017). We found one instance where the pipeline incorrectly flagged a relatively simple website as being too complex (i.e. having too many elements) in Figure 13 (second screen from the bottom), likely because it evaluated the com-

---

[3]https://mobbin.com/
[4]https://designvault.io/

plexity based on the mobile standards presented in the few-shot examples. However, the pipeline did correctly critique the bottom screenshot in the same figure for being overly complex, showing that it can appropriately identify this issue in some cases.

### A.5.3 REFINING BOUNDING BOXES

While the bounding boxes could be improved qualitatively, as shown in Figures 5, 6, 7, and 8, there are straightforward approaches to easily improve the bounding box accuracy. For instance, the DOM tree representation of the UI contains the exact bounding boxes of UI elements and element groups. The DOM tree is available through the UI's XML code, or the internal UI mockup representation available in design tools like Figma. If the DOM tree is not available, we could use a screen object parser (Wu et al., 2021; Xie et al., 2020) to extract the exact bounding boxes of UI elements and groups from the screenshot. These exact bounding boxes could then be used to refine the output bounding boxes for the critiques by updating them to precisely contain the exact bounding boxes of the closest UI elements or groups. This matching could be achieved through methods like IoU comparison, computing distances between the bounding box centers and sizes, or utilizing an VLM for matching, as was done by Zhao et al. (2024).

We demonstrate the refinement results from using the UI screenshot element detector created by Xie et al. (2020), which extracts the exact bounding boxes of all UI elements, for a few of the UICrit UIs in Figures 5, 6, 7, and 8. We refined the generated bounding boxes to precisely contain the closest exact UI element bounding boxes, determined via an IoU threshold. Since only the exact bounding boxes for UI elements were available, output bounding boxes intended for UI groups may be refined to include the exact bounding boxes of multiple UI elements. The refined results, shown in Figures 14 and 15, demonstrate that this simple refinement approach, which requires only the UI screenshot as input and no additional data (e.g., the DOM tree), significantly improves bounding box accuracy. This step could potentially be applied at the end of the pipeline to further clean up the generated bounding boxes.

### A.6 ANALYSIS OF ITERATIVE REFINEMENT AND PIPELINE COST ANALYSIS

Figure 16 illustrates an example of iterative bounding box refinement (conditioned on the comment) by BoxRefine, which gradually improves the bounding box and terminates on a significantly more accurate bounding box. Figure 17 illustrates an example of comment refinement (conditioned on the bounding box) by TextRefine, which terminates on an accurate comment on the poor layout of the region inside the bounding box.

We calculated the average number of bounding box refinements, which were 1.25 for Gemini-1.5-pro and 0.88 for GPT-4o, as well as the average number of comment refinements, which were 1.48 for Gemini-1.5-pro and 1.17 for GPT-4o. Additionally, we estimated the expected number of VLM calls required for a complete run of the pipeline, including the small fraction sent for further refinement by Validation. The expected number of calls is 7.16 for Gemini-1.5-pro and 6.70 for GPT-4o.

### A.7 HUMAN EVALUATION METHOD

Figure 18 shows a snippet of the form used by human design experts to rate the quality of individual comments and rank the comment sets for the three different conditions.

Given the limited availability of UI design experts and the extensive evaluation required per UI screen for a detailed comparison across the three conditions, only the Gemini-1.5-pro outputs for 33 UI screenshots were rated. To better represent the UI design space in this sample, we maximized the diversity of the UI screenshots by randomly sampling an even number of UIs from each of the UI task categories identified by Duan et al. (2024a). We followed their method of clustering by task descriptions from UICrit to obtain the task clusters. These 33 UIs were split into 6 groups for rating, with three participants assigned to each group. The rating and ranking study took approximately 1 hour. We recruited 18 design experts for this study. Five of the participants had 2-4 years of design experience, and the rest had 6-10 years. Their areas of design expertise include mobile, web, interaction, and user experience research.

## A.8 Algorithms for Generating Few-shot Examples for Bounding Box Refinement

Algorithm 1 details the steps for generating the few-shot refinement examples for a selected bounding box. The few-shot generation algorithm entails perturbing the bounding box coordinates by decreasing amounts and adding the perturbations to the list of few-shot examples. The algorithm for perturbing a bounding box is also shown in Algorithm 1.

---

**Algorithm 1** Generate Bounding Box Refinement Few-shot Examples

---

**Require:** the bounding box to be perturbed $input\_bbox$, the fraction that the bounding box's coordinates will be perturbed $perturb\_frac$

**Ensure:** The coordinates of $input\_bbox$ perturbed by $perturb\_frac$

1: **function** GENERATE_PERTURB($input\_bbox$, $perturb\_frac$)
2:     Compute $left\_margin$, $right\_margin$, $top\_margin$, $bottom\_margin$
3:     $all\_perturbed \leftarrow []$
4:     **for** $x\_perturb$ in $[-perturb\_frac \times left\_margin, perturb\_frac \times right\_margin]$ **do**
5:         **for** $y\_perturb$ in $[-perturb\_frac \times top\_margin, perturb\_frac \times bottom\_margin]$ **do**
6:             Update bounding box location based on $x\_perturb$, $y\_perturb$
7:             Add perturbed bounding box to $all\_perturbed$
8:         **end for**
9:     **end for**
10:    $final\_perturbed \leftarrow []$
11:    Compute $width$ and $height$ of the input bounding box
12:    **for** each $perturbed\_bbox$ in $all\_perturbed$ **do**
13:        **for** $width\_fraction$ in $[-perturb\_frac, perturb\_frac]$ **do**
14:           **for** $height\_fraction$ in $[-perturb\_frac, perturb\_frac]$ **do**
15:              Update bounding box size based on $width\_fraction$ and $height\_fraction$
16:           **end for**
17:        **end for**
18:    **end for**
19:    $filtered\_perturbed \leftarrow remove\_invalid\_perturbed\_bbox(final\_perturbed, input\_bbox)$
20:    $final\_bbox \leftarrow random.choice(filtered\_perturbed)$
21:    **return** $final\_bbox$
22: **end function**

---

**Require:** Bounding box $bbox$, maximum number of perturbations of $bbox$ in the list of fewshot refinement examples $max\_num\_perturb$

**Ensure:** A list of bounding boxes coordinates that are perturbed versions of $bbox$ in decreasing amounts, where $bbox$ is the last item in the list.

23: **function** GENERATE_PERTURBED_FEWSHOT_EXAMPLES($bbox$, $max\_num\_perturb$)
24:    $perturb\_options \leftarrow$ LIST(range($max\_num\_perturb + 1$))
25:    $num\_perturb \leftarrow$ RANDOM_CHOICE($perturb\_options$)
26:    $perturb\_list \leftarrow []$
27:    **for** $j \leftarrow num\_perturb$ **to** 1 **do**
28:        $perturb\_frac \leftarrow j/max\_num\_perturb$
29:        $output\_bbox \leftarrow$ GENERATE_PERTURB($bbox$, $perturb\_frac$)
30:        $perturb\_list$.append($output\_bbox$)
31:    **end for**
32:    $perturb\_list$.append($bbox$)
33:    **return** $perturb\_list$
34: **end function**

---

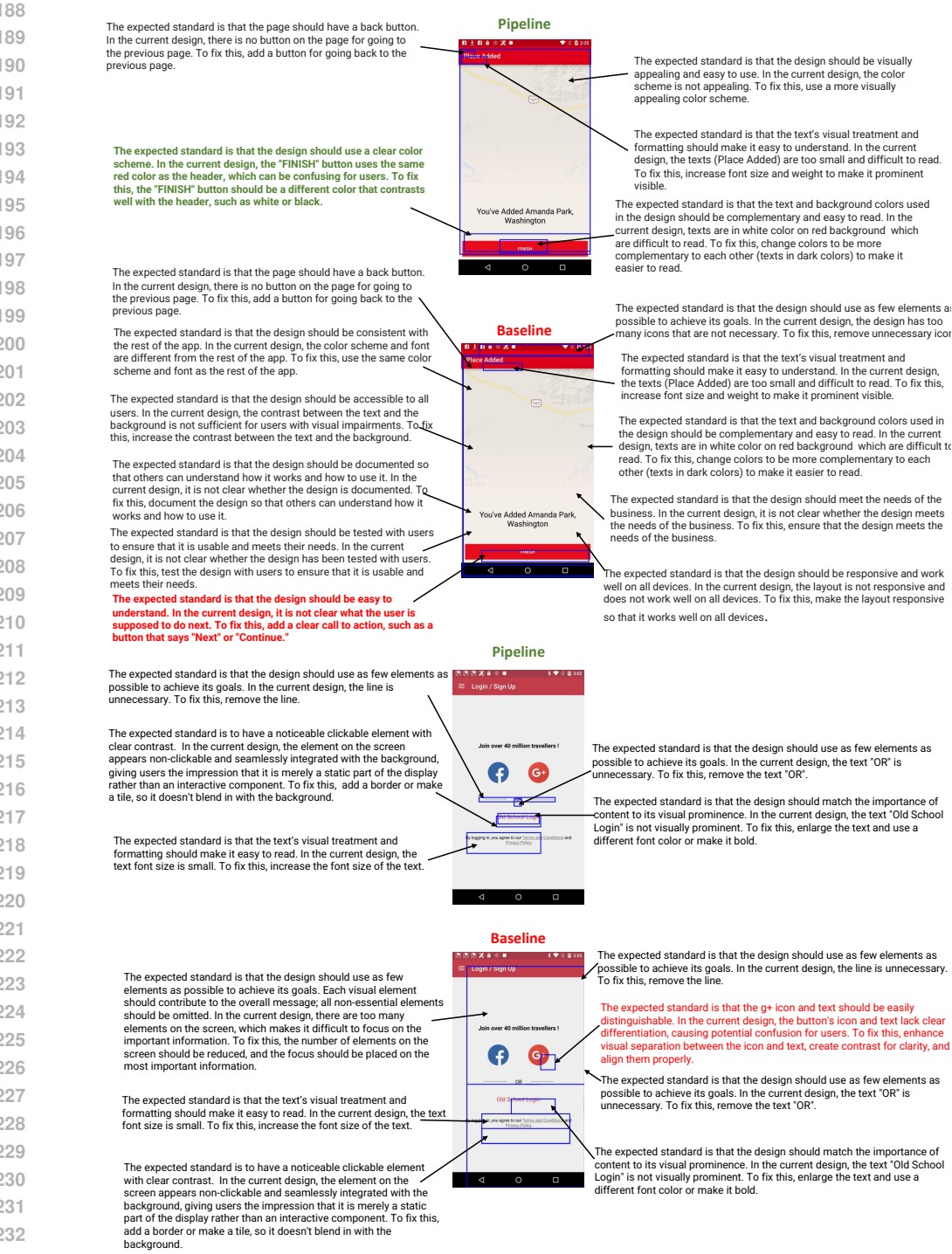

Figure 8: Illustration of outputs from the pipeline and baseline, highlighting two cases where our pipeline outperformed the baseline. The screenshots are marked with the output bounding boxes, and the generated comments are shown, each pointing to its corresponding bounding box (some comments have the same bounding box). Both the pipeline and baseline begin with the TextGen module, so we used the same initial comments from TextGen for both conditions for easier comparison. In the top example, the pipeline produced more accurate bounding boxes, eliminated several generic and unhelpful comments, and refined an inaccurate comment (red) into a more accurate one (green). In the bottom example, the pipeline produced more considerably more accurate bounding boxes, and eliminated an invalid comment (red).

**Pipeline**

The expected standard is that the design should be consistent throughout. In the current design, the share icon is not consistent. To fix this, we can use the same share icon.

**Baseline**

The expected standard is that the design should be consistent throughout. In the current design, the share icon is not consistent. To fix this, we can use the same share icon.

The expected standard is that the design should be consistent throughout. In the current design, the three dot icon is not consistent. To fix this, we can use the same three dot icon.

The expected standard is that the design should be consistent throughout. In the current design, the date format is not consistent. To fix this, we can use the same date format.

The expected standard is that the text's visual treatment and formatting should make it easy to read. In the current design, the text font size is small. To fix this, we can increase the text font size.

**Pipeline**

The expected standard is that the design should be visually appealing and easy to use. In the current design, the font size of the text "Round 1/12" is too small. To fix this, the designer should increase the font size of the text "Round 1/12

The expected standard is that the text's visual treatment and formatting should make it easy to read. In the current design, the text font size is small and the background makes the foreground text difficult to read. To fix this, we can increase the text font size and choose a different contrasting background.

The expected standard is that the text's visual treatment and formatting should make it easy to read. In the current design, the text font size is small and the background makes the foreground text difficult to read. To fix this, we can increase the text font size and choose a different contrasting background.

The expected standard is that the design should use as few elements as possible to achieve its goals. Each visual element should contribute to the overall message; all non-essential elements should be omitted. In the current design, there are too many elements on the screen, making it difficult to focus on any one thing. To fix this, the designer should remove any unnecessary elements from the screen.

The expected standard is that the design should be visually appealing and easy to use. In the current design, the font size of the text "Round 1/12" is too small. To fix this, the designer should increase the font size of the text "Round 1/12

**Baseline**

The expected standard is that the design should match the importance of content to its visual prominence. In the current design, the text "classic boxing" is not visually prominent. To fix this, we can increase the font size of the text "classic boxing".

The expected standard is that the design should be visually appealing and easy to use. In the current design, the text is not aligned properly. To fix this, the designer should align the text to the center.

The expected standard is that the design should be visually appealing and easy to use. In the current design, the spacing between the elements is not consistent. To fix this, the designer should use a consistent spacing between the elements.

The expected standard is that the design should match the importance of content to its visual prominence. In the current design, the text within the highlighted buttons lacks visual prominence. To fix this, we can increase the text font size.

The expected standard is that the design should be visually appealing and easy to use. In the current design, the buttons are too close to each other. To fix this, the designer should add more space between the buttons.

The expected standard is that the design should match the importance of content to its visual prominence. In the current design, the download button is not visually prominent. To fix this, we can enlarge the download button.

The expected standard is that the design should use as few elements as possible to achieve its goals. Each visual element should contribute to the overall message; all non-essential elements should be omitted. In the current design, there are too many elements on the screen, making it difficult to focus on any one thing. To fix this, the designer should remove any unnecessary elements from the screen.

The expected standard is that the design should be visually appealing and easy to use. In the current design, the font size of the text "Round 1/12" is too small. To fix this, the designer should increase the font size of the text "Round 1/12

Figure 9: Illustration of outputs from the pipeline and baseline, highlighting two cases where the baseline outperformed our pipeline. The screenshots are marked with the output bounding boxes, and the generated comments are shown, each pointing to its corresponding bounding box (some comments have the same bounding box). Both the pipeline and baseline begin with the TextGen module, so we used the same initial comments from TextGen for both conditions for easier comparison. For the top example, while a lot of the comments from the baseline are inaccurate, the pipeline eliminated the only correct comment (green) and only kept an invalid comment (red), though its bounding box is considerably more accurate. In the bottom example, the pipeline removed two valid comments (green) and some invalid ones, and also made the bounding box around the comment regarding the red buttons less accurate.

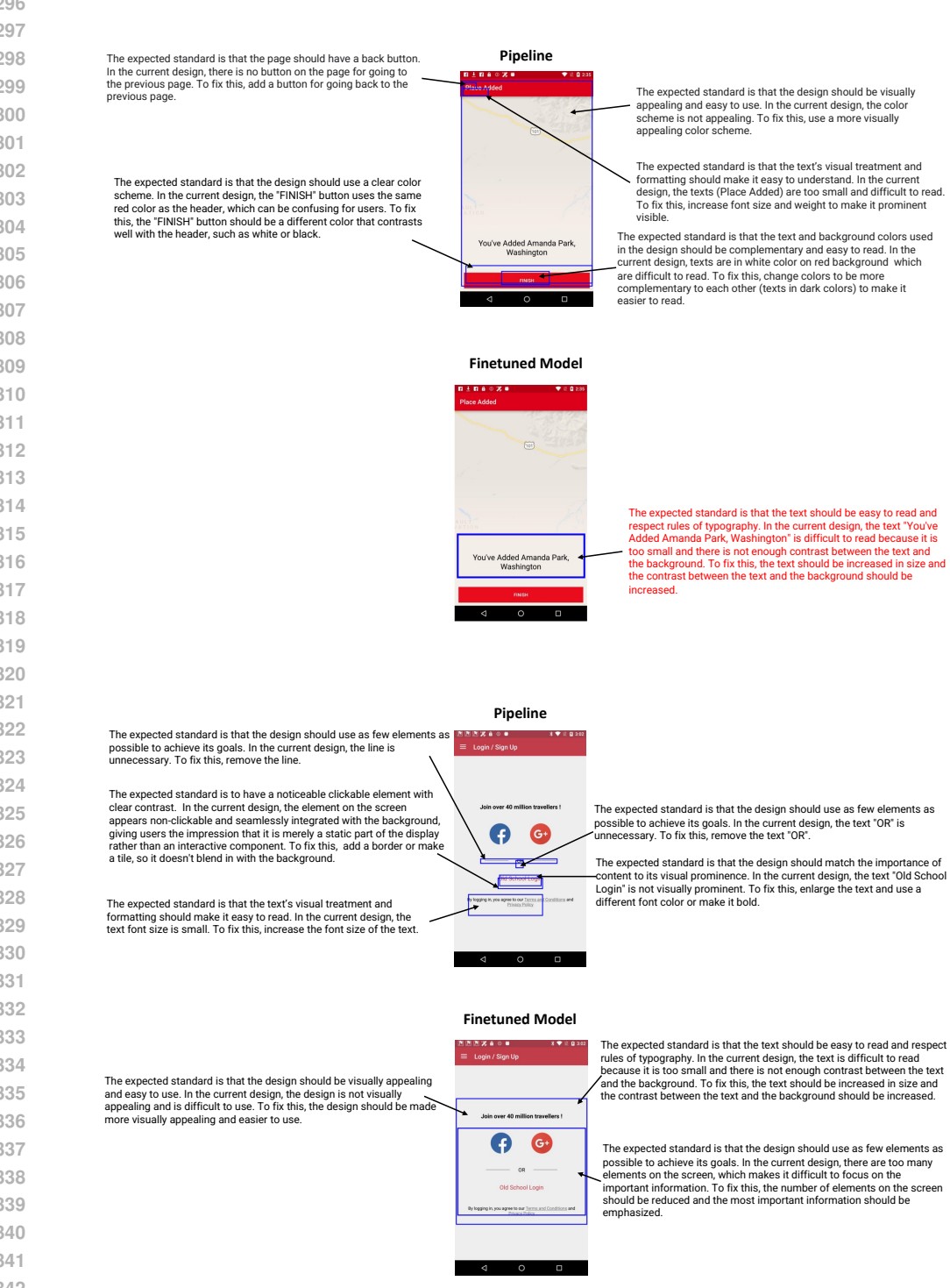

Figure 10: Illustration of outputs from the pipeline and finetuned Llama-3.2 11b. The screenshots are marked with the output bounding boxes, and the generated comments are shown, each pointing to its corresponding bounding box (some comments have the same bounding box). The fine-tuned model produces a limited range of critiques, some of which are inaccurate (red), though the bounding boxes are generally accurate. In contrast, the pipeline generates a significantly more diverse set of critiques, and its bounding boxes are tighter but generally less accurate.

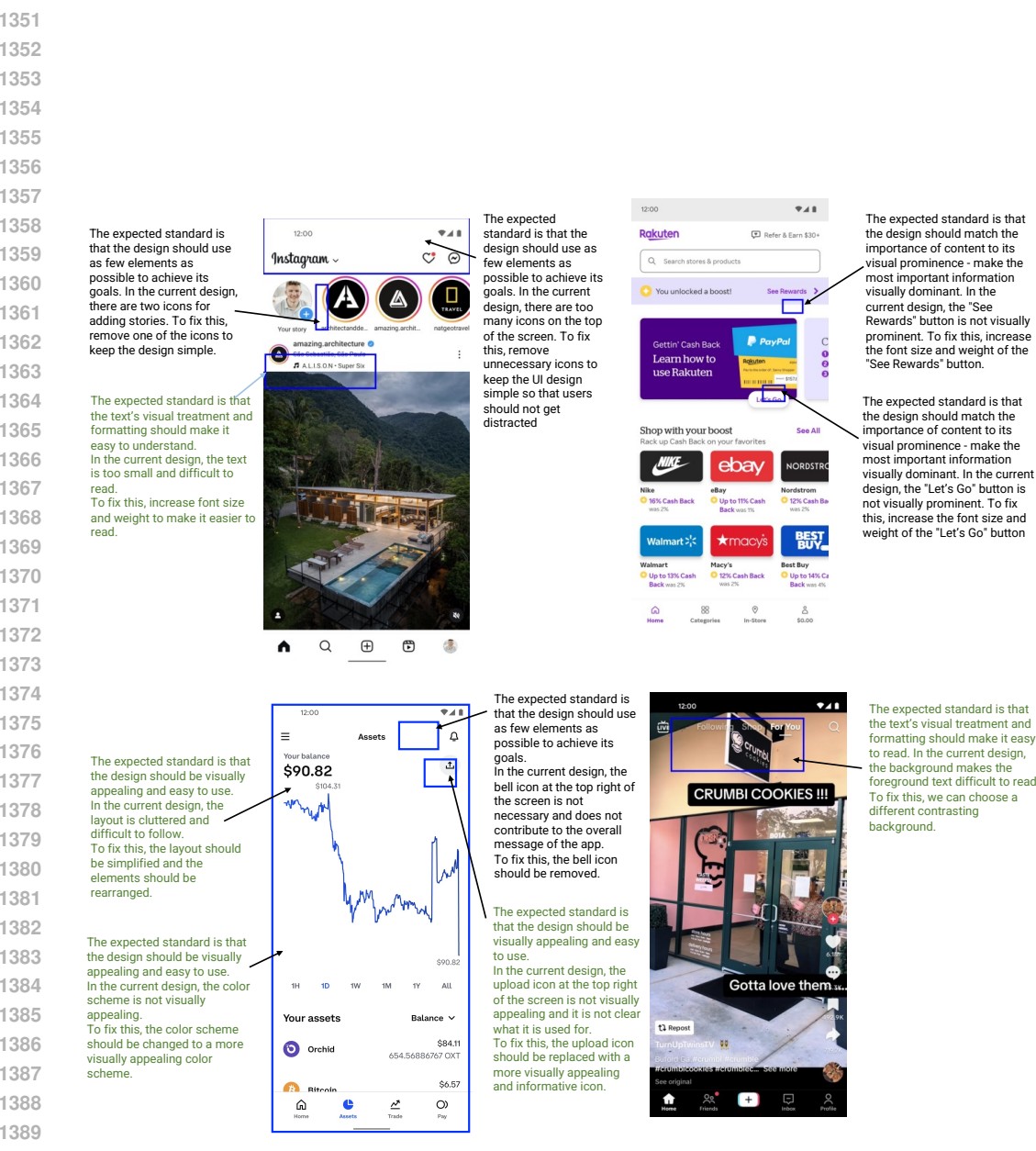

Figure 11: Example design feedback and bounding boxes generated by our pipeline for four modern Android UIs (from 2024). These UIs are out-of-domain inputs, as we used fewshot examples from only UICrit, which consists of older UIs (from 2014). The screenshots are marked with the output bounding boxes, and the generated comments are shown, each pointing to its corresponding bounding box. Helpful comments with reasonably accurate bounding boxes are highlighted in screen.

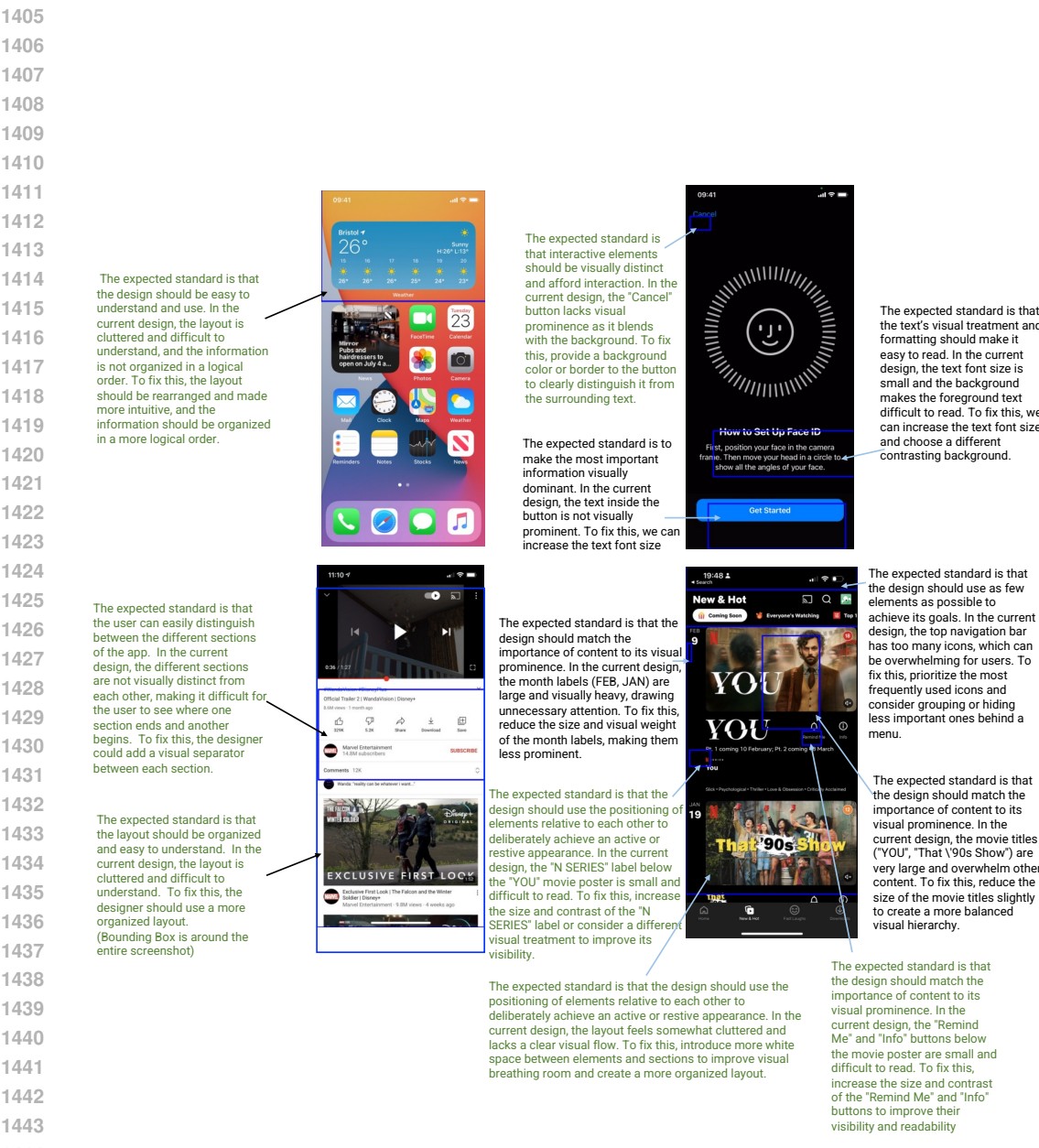

The expected standard is that the design should be easy to understand and use. In the current design, the layout is cluttered and difficult to understand, and the information is not organized in a logical order. To fix this, the layout should be rearranged and made more intuitive, and the information should be organized in a more logical order.

The expected standard is that interactive elements should be visually distinct and afford interaction. In the current design, the "Cancel" button lacks visual prominence as it blends with the background. To fix this, provide a background color or border to the button to clearly distinguish it from the surrounding text.

The expected standard is to make the most important information visually dominant. In the current design, the text inside the button is not visually prominent. To fix this, we can increase the text font size

The expected standard is that the text's visual treatment and formatting should make it easy to read. In the current design, the text font size is small and the background makes the foreground text difficult to read. To fix this, we can increase the text font size and choose a different contrasting background.

The expected standard is that the user can easily distinguish between the different sections of the app. In the current design, the different sections are not visually distinct from each other, making it difficult for the user to see where one section ends and another begins. To fix this, the designer could add a visual separator between each section.

The expected standard is that the layout should be organized and easy to understand. In the current design, the layout is cluttered and difficult to understand. To fix this, the designer should use a more organized layout. (Bounding Box is around the entire screenshot)

The expected standard is that the design should match the importance of content to its visual prominence. In the current design, the month labels (FEB, JAN) are large and visually heavy, drawing unnecessary attention. To fix this, reduce the size and visual weight of the month labels, making them less prominent.

The expected standard is that the design should use the positioning of elements relative to each other to deliberately achieve an active or restive appearance. In the current design, the "N SERIES" label below the "YOU" movie poster is small and difficult to read. To fix this, increase the size and contrast of the "N SERIES" label or consider a different visual treatment to improve its visibility.

The expected standard is that the design should use the positioning of elements relative to each other to deliberately achieve an active or restive appearance. In the current design, the layout feels somewhat cluttered and lacks a clear visual flow. To fix this, introduce more white space between elements and sections to improve visual breathing room and create a more organized layout.

The expected standard is that the design should use as few elements as possible to achieve its goals. In the current design, the top navigation bar has too many icons, which can be overwhelming for users. To fix this, prioritize the most frequently used icons and consider grouping or hiding less important ones behind a menu.

The expected standard is that the design should match the importance of content to its visual prominence. In the current design, the movie titles ("YOU", "That \'90s Show") are very large and overwhelm other content. To fix this, reduce the size of the movie titles slightly to create a more balanced visual hierarchy.

The expected standard is that the design should match the importance of content to its visual prominence. In the current design, the "Remind Me" and "Info" buttons below the movie poster are small and difficult to read. To fix this, increase the size and contrast of the "Remind Me" and "Info" buttons to improve their visibility and readability

Figure 12: Example design feedback and bounding boxes generated by our pipeline for four modern iOS UIs (from 2024). These UIs are out-of-domain inputs, as we used fewshot examples from only UICrit, which consists of older UIs (from 2014). The screenshots are marked with the output bounding boxes, and the generated comments are shown, each pointing to its corresponding bounding box. Helpful comments with reasonably accurate bounding boxes are highlighted in screen.

The expected standard is to make the most important information visually dominant. In the current design, the icons are not visually prominent. To fix this, we can enlarge the icons

The expected standard is that the text's visual treatment and formatting should make it easy to read. In the current design, the text font size is small. To fix this, we can increase the text font size.'

The expected standard is that the design should use clear and concise language. In the current design, some of the language used is jargonistic and could be difficult for users to understand. To fix this, the designer should use clear and concise language that is easy for the user to understand.

The expected standard is that the text's visual treatment and formatting should make it easy to read. In the current design, the text font size for "Per user / month, billed yearly" is small and difficult to read. To fix this, we can increase the text font size

The expected standard is that the design should use as few elements as possible to achieve its goals, and each visual element should contribute to the overall message. In the current design, there are too many elements on the page and it is difficult to focus on the most important information. To fix this, the design should be simplified and the number of elements should be reduced.

The expected standard is that the design should be consistent in its use of typography, color, and layout. In the current design, the use of typography, color, and layout is inconsistent. To fix this, the designer should create a style guide that defines the typography, color palette, and layout grid for the design.

The expected standard is that the text's visual treatment and formatting should make it easy to read. In the current design, the text is too small and difficult to read. To fix this, increase the font size.

The expected standard is to make the most important information visually dominant. In the current design, the highlighted text is small and difficult to read. To fix this, we can increase the text font size.

The expected standard is to make the most important information visually dominant. In the current design, the close button is not visually prominent. To fix this, we can enlarge the close button.

The expected standard is that the design should use a consistent visual hierarchy to distinguish between different levels of importance in the content. In the current design, the visual hierarchy is not clear, and it is difficult to distinguish between different levels of importance in the content. To fix this, use a consistent visual hierarchy to distinguish between different levels of importance in the content.

The expected standard is that the design should use as few elements as possible to achieve its goals. Each visual element should contribute to the overall message; all non-essential elements should be omitted. In the current design, the top navigation bar has too many options, which might overwhelm the user. To fix this, try removing content that does not help convey the primary message

The expected standard is that the design should make the most important information visually dominant. In the current design, the button "Talk to Sales" is not visually prominent. To fix this, we can enlarge the button "Talk to Sales".

The expected standard is that the design should use a clear and easy-to-understand visual hierarchy to help users navigate the interface.
In the current design, the visual hierarchy is unclear and confusing.
To fix this, the designer should use a more distinct visual hierarchy to emphasize the most important elements of the interface. This can be done by using larger font sizes, bolder colors, or more white space.

The expected standard is that the design should be visually appealing and engaging. In the current design, the design is cluttered and overwhelming.
To fix this, the designer should simplify the design and use more white space.

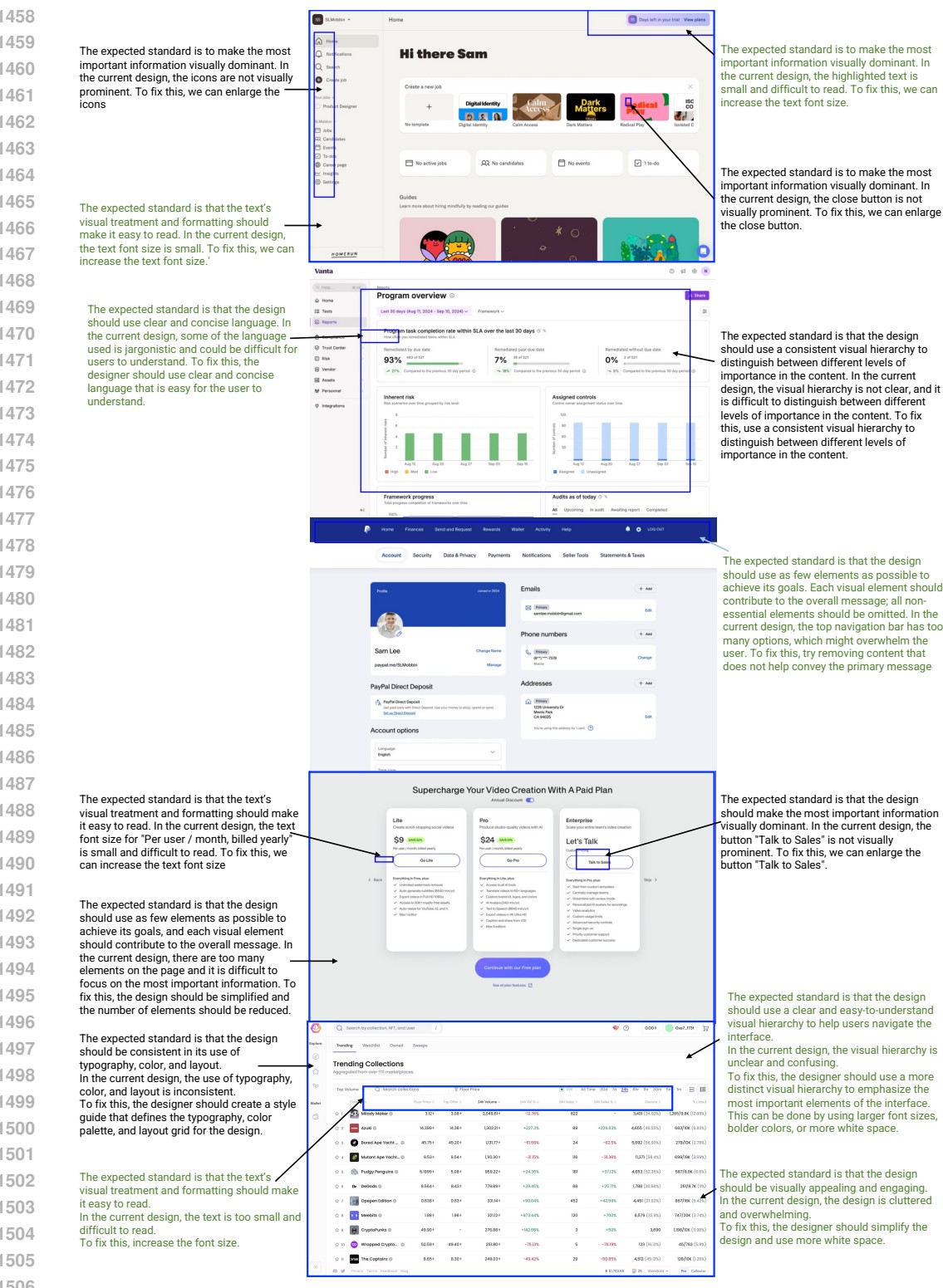

Figure 13: Example design feedback and bounding boxes generated by our pipeline for five modern websites (from 2024). These websites are out-of-domain inputs, as we used fewshot examples from only UICrit, which consists of older mobile UIs (from 2014) that differ significantly from modern websites. The screenshots are marked with the output bounding boxes, and the generated comments are shown, each pointing to its corresponding bounding box. Helpful comments with reasonably accurate bounding boxes are highlighted in screen.

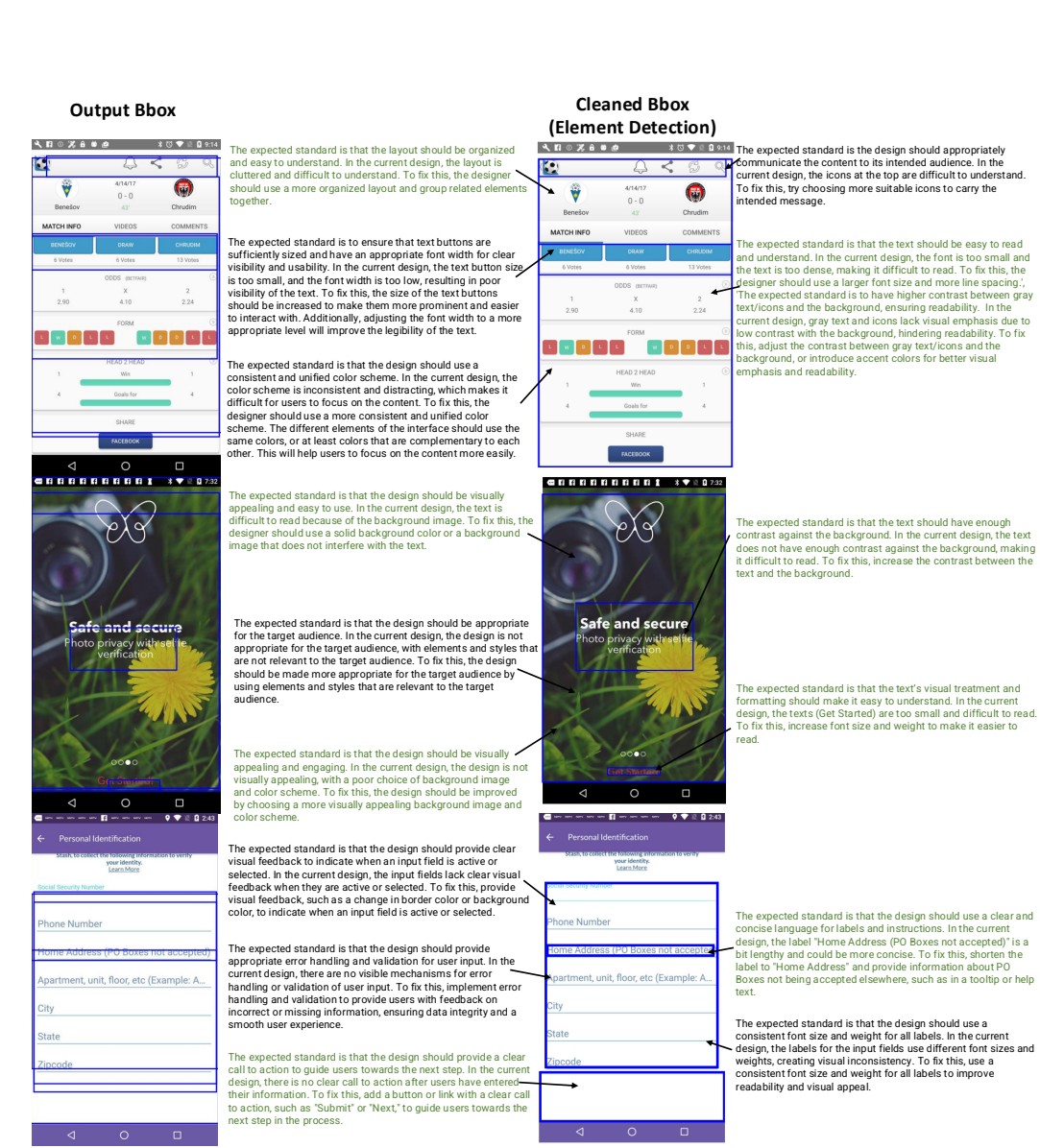

Figure 14: Side by side comparison of the bounding boxes generated by the pipeline ("Output Bbox") and refined bounding boxes ("Cleaned Bbox (Element Detection)"), which were adjusted using the exact bounding boxes of the nearest UI elements. The exact bounding boxes were computed using a UI element detector (Xie et al., 2020), and the nearest elements were determined based on an IoU threshold with the output bounding box. This refinement approach significantly improves the quality of the generated bounding boxes

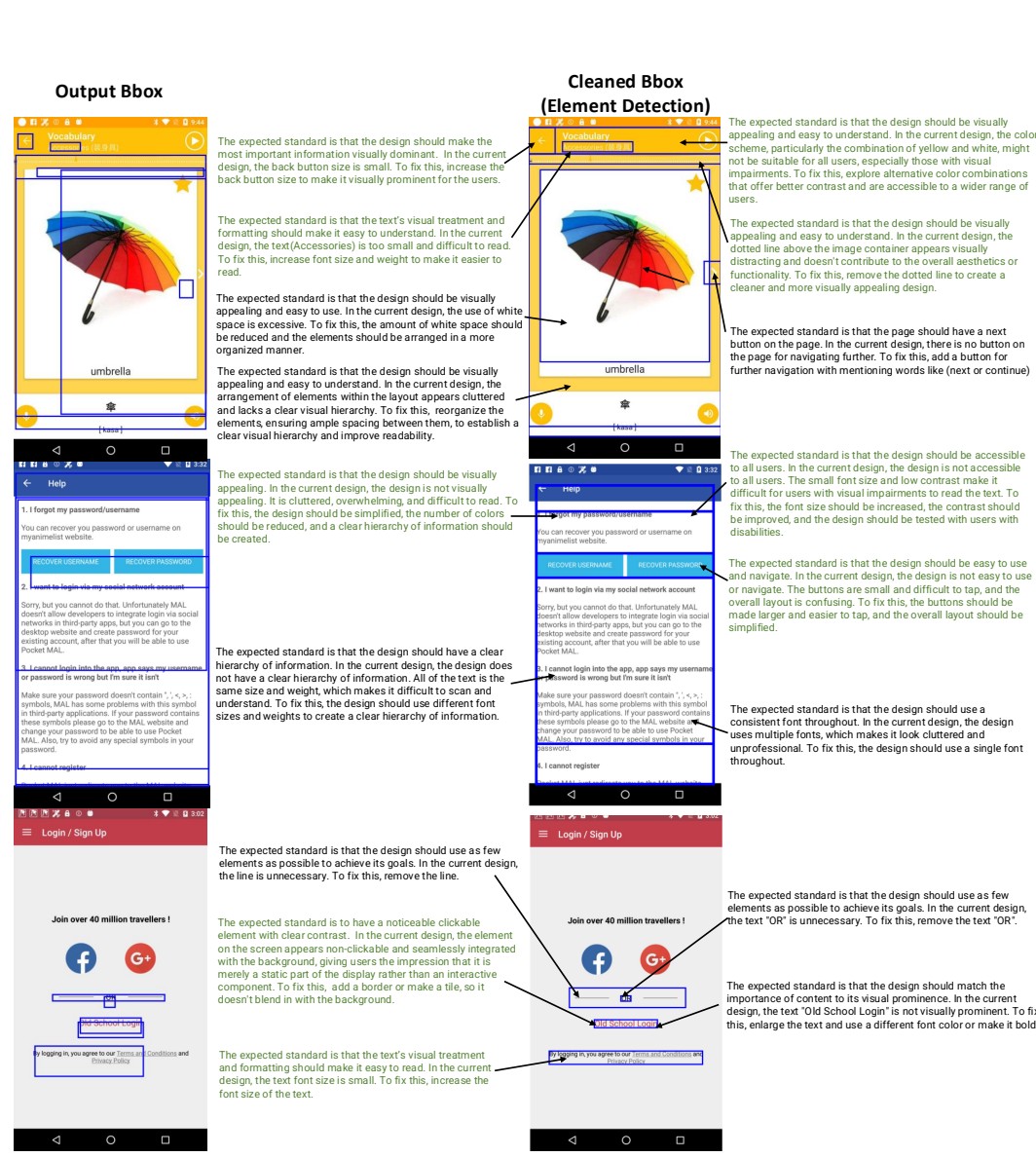

Figure 15: Side by side comparison of the bounding boxes generated by the pipeline ("Output Bbox") and refined bounding boxes ("Cleaned Bbox (Element Detection)"), which were adjusted using the exact bounding boxes of the nearest UI elements. The exact bounding boxes were computed using a UI element detector (Xie et al., 2020), and the nearest elements were determined based on an IoU threshold with the output bounding box. This refinement approach significantly improves the quality of the generated bounding boxes

Start                                           End

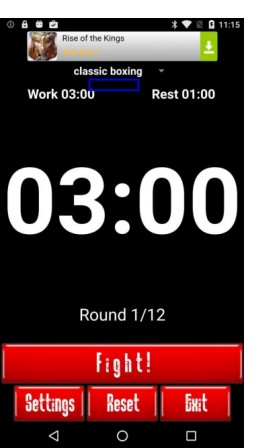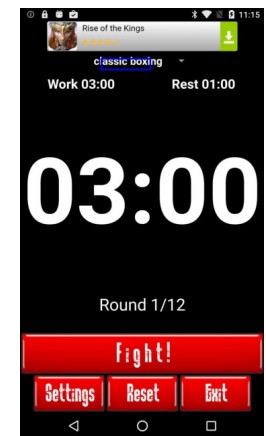

**Comment:** The expected standard is that the design should match the importance of content to its visual prominence. In the current design, the text "classic boxing" is not visually prominent. To fix this, we can increase the font size of the text "classic boxing".

Figure 16: An example of iterative bounding box refinement, with the comment it is conditioned on displayed on the right. The bounding box in the first screenshot ('Start') is the output from BoxGen. The refinement process progressively improves the bounding box, terminating on a significantly more accurate bounding box ('End').

Start                              End                              Screenshot

| Start | End | Screenshot |
|---|---|---|
| The expected standard is that design should convey a clear message In the current design, it does not provide enough information to the users to understand what the app itself is all about. To fix this, redesign it by adding additional information with features to communicate the content to its intended users. | The expected standard is that the design should be consistent throughout the app. In the current design, the "Ringtone" section and the "Message Notification Sounds" section are not consistent with each other. The "Ringtone" section has a dropdown menu, while the "Message Notification Sounds" section does not. To fix this, the designer should make the two sections consistent with each other. For example, both sections could have dropdown menus. | The expected standard is that the design should be easy to understand and use. In the current design, the layout of the notification settings is confusing and difficult to follow. To fix this, the designer should reorganize the layout to make it more intuitive. |

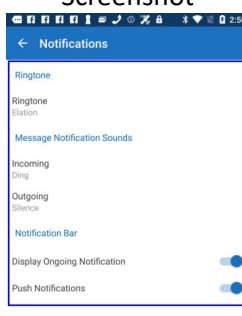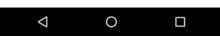

Figure 17: An example of iterative comment refinement, with the bounding box it is conditioned on displayed on the right. The first comment ('Start') was classified as incorrect by the Validation VLM but has a bounding box that accurately encloses a problematic region in the UI. The refinement process progressively improves the comment, terminating with a comment that correctly points out the poor layout of the region in the bounding box. ('End').

Figure 18: The form used for individual comment quality rating and comment set ranking.

