# OpenReview forum: "Visual Prompting with Iterative Refinement for Design Critique Generation"
_ICLR.cc/2026/Conference — Submitted to ICLR 2026_

### Official Review · Reviewer_y32e · 2025-10-15

**Soundness:** 3
**Presentation:** 2
**Contribution:** 2
**Rating:** 6
**Confidence:** 3

**Summary:**

This paper presents a new system for automating design critiques, primarily targeting user interface design.  The method combines visual prompting and iterative refinement to create a design critique visually grounded by bounding boxes around problematic UI elements.

The method does not require additional training and is evaluated with two frontier VLMs, Gemini-1.5-pro and GPT-4o.   Quantitative results show improvements over a baseline method (Duan et al.) using a human evaluation.  Ablation studies validate each step of the proposed pipeline.  In addition, the iterative refinement of the bounding box prediction sub-task is shown to generalize to an open vocabulary object detection benchmark.

Extensive qualitative results in the appendix show the performance of the technique and include comparisons with the baseline method.

**Strengths:**

The method is sound and coherent.  Each step in the pipeline makes good sense and ablation studies also validate these gains.

A human evaluation of the design critiques is performed.  This is currently the best way measure the performance for functionality of this kind.

The paper shows improved performance over baseline methods both qualitatively and quantitatively.

The proposed method does not require fine tuning and consequently can capitalize on improvements in frontier VLMs, which is a rapidly advancing technology.

**Weaknesses:**

While this is a solid piece of work, the gains are made by relatively obvious extensions of existing techniques.  For example, extending  iterative refinement [Madaan et al. (2023) and Xu et al. (2024a)] to bounding box prediction.

It’s a little hard to tell how much the gains in results would really assist a UI designer.  The quantitative gains appear relatively small, although it's hard to assess the scale of the numbers.  The qualitative comparisons with the baseline method in the appendix were helpful, but without seeing a large number of such comparisons picked at random it’s very hard to know if the performance gain would be noticeable.

The system is passed a list of “Design Guidelines” in addition to an image of the UI.   It's not clear whether these have any impact on the output.  I didn't see any evaluation of this.

The entire system is relatively complex.   These days I become nervous that the next generation of VLMs will be able to get comparable performance from a single prompt without the additional complexity.

**Questions:**

What does \subscript{tn} stand for in the table column headings?   Is it just an abbreviation for -1.5-pro and -4o?

The cost analysis in appendix A6 didn’t give me any feeling for how much more expensive the pipeline would be to run over the Duan et al. baseline method.   I would be interested to understand if the method was more expensive to run.   I see you make 6 or 7 VLM/LLM calls, but I'm not sure how many input and output tokens were used and how that compares to the baseline.

In Figure 1, the "Design Guidelines" section is to small too read without extreme zoom.   The output "Design comments" in the figure do not align with anything I could see in the “Design Guidelines”.   Do these guidelines have any impact at all?  I didn't see any metrics for this.  What happens if you add a very striking guideline like "My brand requires all text in the app to be in glowing neon purple".

---

### Official Review · Reviewer_yrbX · 2025-11-02

**Soundness:** 2
**Presentation:** 3
**Contribution:** 1
**Rating:** 2
**Confidence:** 3

**Summary:**

This paper presents a prompt-based VLM pipeline for automated UI design critique generation without model training, using iterative refinement and visual prompting. ​ It improves over the baseline in comment quality and bounding box accuracy for the UI critique task and generalizes to image object grounding tasks, outperforming the baseline in visual grounding accuracy.

**Strengths:**

- A VLM pipeline improves single VLM for UI design judgement.
- Both visual and textural validation modules improve accuracy.
- Shows generalization to other visual grounding tasks.

**Weaknesses:**

- The method and idea are simple, with limited novelty. No model training is done. No new dataset is involved.
- The method is not tailored much to the target problem, except for a few in context samples used in prompts.
- The critique results is still far from human expert.

**Questions:**

The authors are encouraged to study more UI design principle themselves to better understand the challenges and have more specialized design for the problem.
Need higher quality human rating to judge the results.

---

### Official Review · Reviewer_xXTv · 2025-11-03

**Soundness:** 2
**Presentation:** 2
**Contribution:** 2
**Rating:** 2
**Confidence:** 4

**Summary:**

This paper proposes a multi-stage pipeline for critique generation in UI design feedback. The method decomposes the task into multiple visual-language model (VLM) modules that iteratively generate and refine textual critiques and their corresponding bounding boxes (bboxes) grounded on UI screenshots. The proposed system includes text generation, text filtering, bbox generation and refinement, validation, and text refinement steps, each handled by separate VLMs to avoid self-bias.

The paper introduces a new dataset, UICrit, containing ~11K human-annotated critiques and bounding boxes. Evaluation combines IoU for visual grounding, comment similarity using sentence-BERT embeddings, and human expert evaluation of critique validity and set ranking. Results show incremental improvements over a previous pipeline baseline (Duan et al., 2024a), and the authors also demonstrate limited transfer to OVAD/OVD tasks.

**Strengths:**

**1. Novel Task Formulation**

The paper clearly defines a spatially grounded critique generation task, integrating both text feedback and visual localization.

**2. Systematic Pipeline Design**

Modular decomposition (generation, filtering, refinement) and VLM-based iterative feedback is conceptually clean and potentially extensible.

**3. Dataset Contribution**

The UICrit dataset, with paired text–bbox annotations, could be useful for future multimodal critique or visual reasoning research.

**Weaknesses:**

**1. Limited Empirical Depth**

Experiments are restricted to a small set of baselines and models (Gemini-1.5-Pro, GPT-4o).
The ablations are shallow; there is no analysis of failure cases, generalization across domains, or robustness.

**2. Marginal Quantitative Gains**

Although IoU and similarity scores increase slightly with each module, absolute performance remains low (e.g., IoU < 0.36).
Human evaluation improvements are modest and may not be statistically significant.

**3. Overly Complex but Technically Shallow**

Despite multiple modules, each stage is implemented via prompting rather than genuine model innovation.
The approach is more of a prompt engineering pipeline than a novel algorithmic contribution.

**4. Evaluation Ambiguity**

Comment similarity (cosine on sentence embeddings) is not a reliable proxy for critique quality.
Human evaluation lacks sufficient detail: number of annotators, inter-rater reliability, and consistency are missing.

**Questions:**

See weakness

---

### Meta-Review · Area_Chair_h62o · 2026-01-07

**Summary:**

This paper proposes a prompt-based VLM pipeline for generating visually grounded UI design critiques using iterative refinement of both text and bounding boxes. The approach decomposes the task into multiple VLM-driven stages and introduces a dataset with paired critique text and bounding boxes. Reviewers agree the task is well-defined, the system is coherent, and the results consistently improve over the prior baseline. However, the majority of reviewers raised concerns about limited novelty, modest empirical gains, and evaluation depth. The approach relies entirely on prompt engineering without model or algorithmic innovation, and quantitative improvements remain small with low absolute performance. Human evaluation is seen as appropriate but insufficiently detailed, and the practical impact for real UI designers remains unclear. Overall sentiment is mixed, with two reviewers recommending rejection due to limited contribution, and one reviewer giving a weak accept while noting similar concerns.

**Reviewer Concerns:**

Major concerns:
1/ The pipeline is largely composed of existing techniques (iterative refinement, visual prompting) applied in a straightforward way.
2/ Improvements over the baseline are incremental, with low absolute IoU and modest human evaluation improvements.
3/ Human evaluation lacks detail (annotators, agreement), and embedding-based similarity metrics are not fully convincing proxies for critique quality.
4/ It is difficult to assess whether the gains meaningfully help UI designers in practice.
5/ Reviewers questioned whether similar performance could be achieved with simpler prompting as VLMs improve.

**Reviewer Scores:**

Reviewer xXTv: Reject (2), confident; concerns about novelty and evaluation.
Reviewer yrbX: Reject (2); views contribution as minimal and method as overly simple.
Reviewer y32e: Weak accept (6); acknowledges soundness and improvements but questions impact and novelty.

No author rebuttal is provided and no discussion follows after the initial reviews. Overall, reviewer scores lean negative.

---

### Decision · Program_Chairs · 2026-01-26

Reject